# Cancer-associated fibroblast-secreted CXCL16 attracts monocytes to promote stroma activation in triple-negative breast cancers

Roni Allaoui[1], Caroline Bergenfelz[1,*], Sofie Mohlin[2,*], Catharina Hagerling[1,3,*], Kiarash Salari[3], Zena Werb[3], Robin L. Anderson[4], Stephen P. Ethier[5], Karin Jirström[6], Sven Påhlman[2], Daniel Bexell[2], Balázs Tahin[7], Martin E. Johansson[1,7], Christer Larsson[2] & Karin Leandersson[1]

Triple-negative (TN) breast cancers (ER$^-$PR$^-$HER2$^-$) are highly metastatic and associated with poor prognosis. Within this subtype, invasive, stroma-rich tumours with infiltration of inflammatory cells are even more aggressive. The effect of myeloid cells on reactive stroma formation in TN breast cancer is largely unknown. Here, we show that primary human monocytes have a survival advantage, proliferate *in vivo* and develop into immunosuppressive myeloid cells expressing the myeloid-derived suppressor cell marker S100A9 only in a TN breast cancer environment. This results in activation of cancer-associated fibroblasts and expression of CXCL16, which we show to be a monocyte chemoattractant. We propose that this migratory feedback loop amplifies the formation of a reactive stroma, contributing to the aggressive phenotype of TN breast tumours. These insights could help select more suitable therapies targeting the stromal component of these tumours, and could aid prediction of drug resistance.

[1] Department of Translational Medicine, Cancer Immunology, Lund University, Malmö 205 02, Sweden. [2] Department of Laboratory Medicine, Translational Cancer Research, Lund University, Lund 223 63, Sweden. [3] Department of Anatomy and the Helen Diller Family Comprehensive Cancer Center, University of California, San Francisco, California 94143-0452, USA. [4] Sir Peter MacCallum Department of Oncology, Peter MacCallum Cancer Centre, The University of Melbourne, Melbourne 8006, Australia. [5] Department of Pathology and Laboratory Medicine, Hollings Cancer Center, Medical University of South Carolina, Charleston, South Carolina 29425, USA. [6] Department of Clinical Sciences Lund, Oncology and Pathology, Lund University, Lund 221 85, Sweden. [7] Department of Translational Medicine, Clinical Pathology, Skånes Universitetssjukhus, Malmö 205 02, Sweden. * These authors contributed equally to this work. Correspondence and requests for materials should be addressed to K.L. (email: Karin.Leandersson@med.lu.se).

Breast cancer is the most common cancer among women today and the prognosis is dependent not only on the stage of disease at detection, but also on the type of breast cancer. Breast cancers can be divided into several subtypes, mainly based on expression of oestrogen receptor (ER), progesterone receptor (PR) and human epidermal growth factor receptor 2 (HER2). Using global gene expression profiling, breast cancers can be further categorized into molecular subtypes including the basal-like and luminal subtypes[1]. Triple-negative breast cancers (ER $^-$ PR $^-$ HER2 $^-$; TNBC) constitute a heterogeneous group of breast cancers that largely coincide with the basal-like subtype. TNBCs are highly metastatic tumours with a poor prognosis and there are few treatment options for patients with these cancers[2]. Infiltration of inflammatory cells or the presence of a stroma with reactive, invasive properties, have been associated with poor prognosis in patients with TNBC[3–5]. Furthering our understanding of the role of the tumour stroma and inflammatory cells in TNBC will help elucidate how the tumour microenvironment may contribute to disease progression, drug resistance or may enable treatments to be tailored to patients more effectively.

The tumour microenvironment is composed of extracellular matrix (ECM) and non-malignant stromal cells including fibroblasts, pericytes, immune cells and endothelial cells. The cells of the tumour microenvironment communicate via soluble mediators or intercellular receptor-ligand interactions. Cancer-associated fibroblasts (CAFs), pericytes and innate immune cells, especially tumour-associated macrophages (TAMs), are the main cell types constituting the tumour stroma. It is generally thought that CAFs are recruited from resident fibroblasts or bone marrow-derived progenitor cells (BMDCs), or trans-differentiated from mesenchymal or tumour-derived cells[6]. These cells are then activated by factors in the tumour microenvironment, such as TGF-β, to become myofibroblasts (αSMA $^+$/vimentin $^+$) that promote invasion and metastasis. How CAFs are recruited and activated is still under intense investigation[7–9].

Monocytes are immune cells of the myeloid lineage that are plastic by nature and can give rise to macrophages, dendritic cells and probably also monocytic-myeloid-derived suppressor cells (MDSCs)[10,11]. Tumour-infiltrating myeloid cells, particularly TAMs and MDSCs negatively affect survival in breast cancer patients[12–16]. This negative effect has been ascribed to their immunosuppressive roles and their effects on tumour cell invasion and angiogenesis[7,17]. Both monocytes and BMDCs can promote metastasis to distant sites[18,19]. We have previously shown that a subpopulation of anti-inflammatory myeloid cells (CD163 $^+$) is present in the tumour stroma of TN breast tumours and is associated with unfavourable clinicopathologic features[4]. However, the effects of myeloid cells on stroma formation in TN breast tumours have not been investigated in detail.

Stroma interactions and the effects on tumour development and progression are complex, and it is therefore important to understand the intricate networks within specific tumour types and the cells of their particular tumour microenvironment[20]. In 2011, Elkabets et al. showed that mouse BMDCs could promote stroma formation in TNBCs, specifically by recruitment or activation of non-bone marrow-derived fibroblasts via secreted granulin (GRN)[19]. Using the same human breast cancer cohort as Elkabets et al., we have shown that the presence of myeloid CD163 $^+$ cells in the stromal areas of human TN breast tumours specifically correlates with these GRN expressing cells[4].

In this study, we investigated whether myeloid cells could affect stroma formation in breast cancer. We show that primary human monocytes, co-transplanted with either luminal A or TNBC cells in highly immunodeficient NSG-mice, differentiated into CD163 $^+$ myeloid cells and promoted an increased stroma formation in both tumour types. However, only the TNBC/monocyte co-transplants developed immunosuppressive CD163 $^+$ myeloid cells, and were able to activate fibroblasts. The findings were validated in patient-derived xenografts (PDXs) as well as syngeneic breast cancer models of luminal and TNBC. Interestingly, we also found that the monocyte chemoattractant CXCL16 was induced in primary fibroblasts cultured under TNBC/monocyte conditions in vitro. In line with these data, primary CAFs isolated from TN tumours specifically expressed CXCL16, a finding that was supported by analysis of a human breast cancer RNAseq data set and a human breast cancer tissue microarray. Our data indicate that in a TNBC environment, myeloid cells can activate the stromal fibroblasts to express CXCL16 that, in turn, recruits more myeloid cells and fibroblasts. On the basis of these findings, we propose that drugs targeted at immunosuppressive myeloid cells, or the circuits mediated by them, such as immune checkpoint inhibitors or immunomodulatory drugs, should be investigated to treat TN breast tumours. Our findings will be important for predicting drug resistance and outcome in patients treated with drugs targeting tumour-infiltrating myeloid cells in the future.

## Results

**Immunosuppressive myeloid cells in TNBC xenografts.** To investigate whether myeloid cells can promote stroma formation in TN tumours, or if the stroma of TN tumours secretes factors that attract CD163 $^+$ myeloid cells, we first generated breast cancer xenograft models. To that end, we co-xenotransplanted primary human monocytes (Mo; the proposed precursors of CD163 $^+$ myeloid cells) from healthy blood donors together with human breast cancer cells of the luminal A (MCF-7 or T47D cells) or TN (MDA-MB-231 or SUM-159 cells) subtypes in highly immunodeficient NSG-mice. These cell lines were chosen since they lack endogenous expression of the myeloid-derived suppressor cell marker, S100A9 (ref. 21), and the NSG-mice were chosen since they are deficient in T-, B- and NK-cells and also have defective macrophages and dendritic cells, but allow engraftment of functional human myeloid cells[22].

Tumours that formed were excised, and their biological characteristics, their myeloid cell content and stroma formation were analysed by immunohistochemistry (IHC) (Supplementary Tables 1 and 2, Figs 1 and 2 and Supplementary Figs 1 and 2). Grafted monocytes (expression of human myeloid markers CD11b and CD163 and very weak expression of human macrophage marker CD68; Fig. 1, Supplementary Tables 1 and 2, Supplementary Figs 1 and 2A) were present in the xenografts at the time of dissection (day 21 for MDA-MB-231, T47D and SUM-159 grafts; day 21 and day 90 for MCF-7 grafts). The presence of CD163 $^+$ or CD68 $^+$ cells was significantly higher in TN as compared with luminal A xenografts (Supplementary Tables 1 and 2). The luminal T47D/monocyte grafts showed the lowest density of CD11b $^+$ cells and lacked CD163 $^+$ cells, whereas the TN SUM-159/monocyte grafts had the highest density of CD163 $^+$ cells (Supplementary Fig. 1). In addition, mouse macrophages (F4/80 staining Supplementary Fig. 2B), human myeloid dendritic cells (CD208) and human fibroblasts (CD90) were all absent (Supplementary Fig. 2B). Gr-1, a mouse myeloid cell marker, was present only at low levels in some xenograft sections (Supplementary Fig. 2C). Of importance, the myeloid-derived suppressor cell marker S100A9 was expressed solely in the myeloid cells that had been co-transplanted with TNBC cells (Supplementary Tables 1 and 2), indicating that these transplanted myeloid cells had acquired an immunosuppressive character[16] (Figs 1, 2a, Supplementary Figs 1 and 2A).

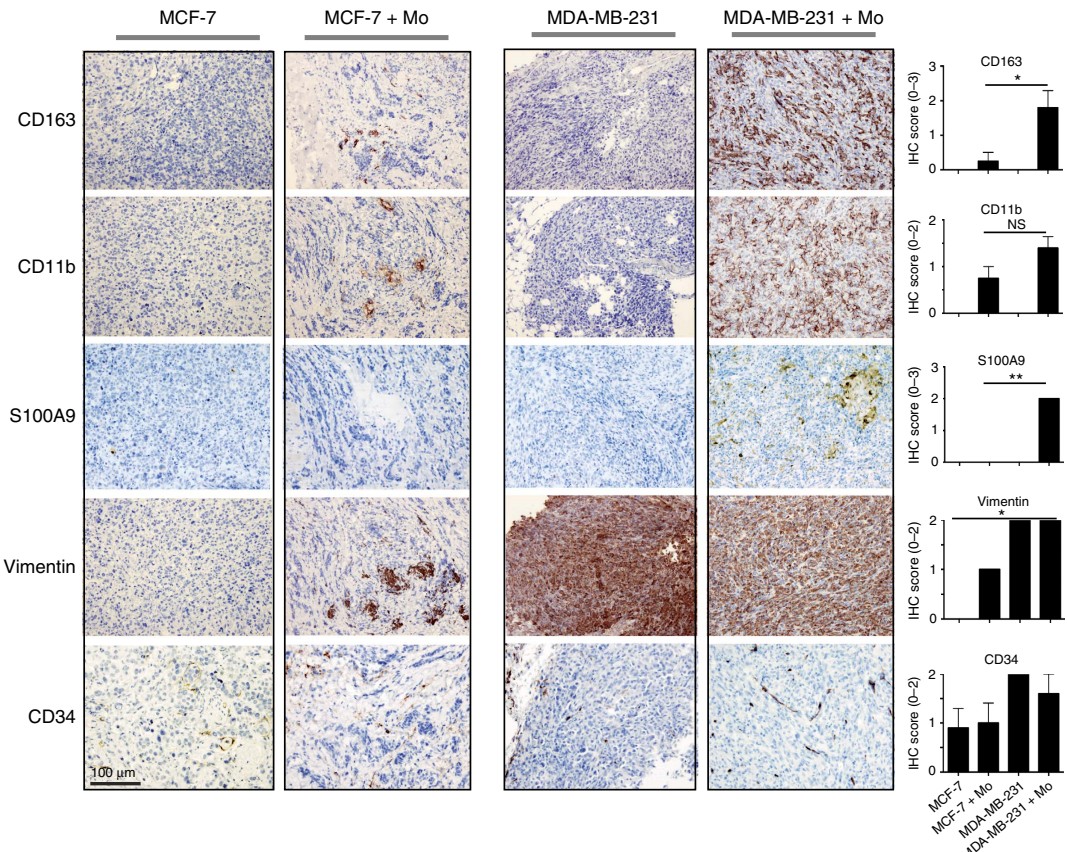

**Figure 1 | IHC of xenografts.** Tumour xenografts consisting of TN MDA-MB-231 breast cancer cells co-transplanted with primary human monocytes, express more myeloid-related and immunosuppressive markers than luminal A MCF-7/monocyte xenografts. The xenografts were grown in highly immunodeficient NSG-mice (see 'Methods' section), and sections from the tumours were stained with myeloid (CD163, CD11b, S100A9) tumour (vimentin) and endothelial markers (CD34). The two cell lines chosen are negative for S100A9 (ref. 21). IHC was performed using the indicated antibodies. All histological sections were counterstained with HE. $N = 5$ mice were analysed for each group; MCF-7 grafts were analysed on day 21 and 90 post-graft with similar results—day 90 is shown here; TN MDA-MB-231 grafts were grown to day 21 only. The histograms show the mean value for each IHC score with statistical analyses. IHC scores are shown in Supplementary Table 1. $* = P < 0.05$, $** = P < 0.01$ ANOVA non-parametric Kruskal–Wallis test. $N = 5$. Error bars indicate s.e.m.

Vimentin is normally expressed by MDA-MB-231 breast cancer cells, but not by MCF-7 cells. In the MCF-7 co-transplants, the vimentin staining is predominantly in the co-transplanted myeloid cells (Fig. 1), as supported by immunofluorescence (Supplementary Fig. 3A), IHC (Supplementary Fig. 3B) and western blotting (Supplementary Fig. 3C). Both MCF-7 and MDA-MB-231 monocyte co-transplants had slightly increased levels of the mouse endothelial marker CD34, suggesting that monocytes promote angiogenesis equally well in luminal A and TN breast xenografts (Fig. 1). Taken together, these data suggest that grafted monocytes survive transplantation in both luminal A and TN breast xenografts. However, myeloid cells were numerous in the TN xenotransplants and expressed the myeloid immuno-suppressive marker, S100A9, only in the TN xenotransplants.

**Monocytes survive and proliferate in a TNBC environment.** We next investigated why the proportion of CD11b$^+$ CD163$^+$ myeloid cells was high in the TN co-transplant tumours. In general, the monocyte co-transplanted tumours were smaller than the corresponding tumour cell-only transplants, and in one case even failed to grow (see Supplementary Tables 1 and 2) with the exception of the TN SUM-159 co-transplants that increased significantly in size in the presence of monocytes (Supplementary Table 2 and Fig. 3a,b). We used the Ki67 proliferation marker to show that the monocytes in the TN, but not the luminal tumours,

were actively proliferating (Fig. 3c). This observation was supported by the fact that primary human monocytes proliferated when cultured in conditioned medium from MDA-MB-231 or SUM-159 cells, but did not proliferate in MCF-7- or T47D-conditioned medium as compared with control medium (Fig. 3d). Conditioned medium from a third TN cell line (MDA-MB-468 cells) did not induce monocyte proliferation (Fig. 3d). To evaluate monocyte survival in different tumour microenvironments, we cultured primary human monocytes in conditioned medium from five different breast cancer cell lines; three TN (MDA-MB-231, MDA-MB-468 and SUM-159) and two luminal A (MCF-7 and T47D). Indeed, survival of monocytes was significantly increased in the TNBC supernatant cultures, as measured by staining with the apoptosis markers annexin V and 7AAD, compared with in the luminal A breast cancer conditioned medium cultures (Fig. 3e and Supplementary Fig. 4A). As we have shown previously[23], both primary human monocytes and M2 macrophages migrated significantly more towards conditioned medium from TN cells than from luminal A cells (Fig. 3f and Supplementary Fig. 4B). Hence, our data indicate that the TN tumour microenvironment promotes increased survival and proliferation of co-transplanted myeloid cells.

**Soluble mediators in monocyte and TNBC cell co-cultures.** To search for soluble factors that might be critical for the increased

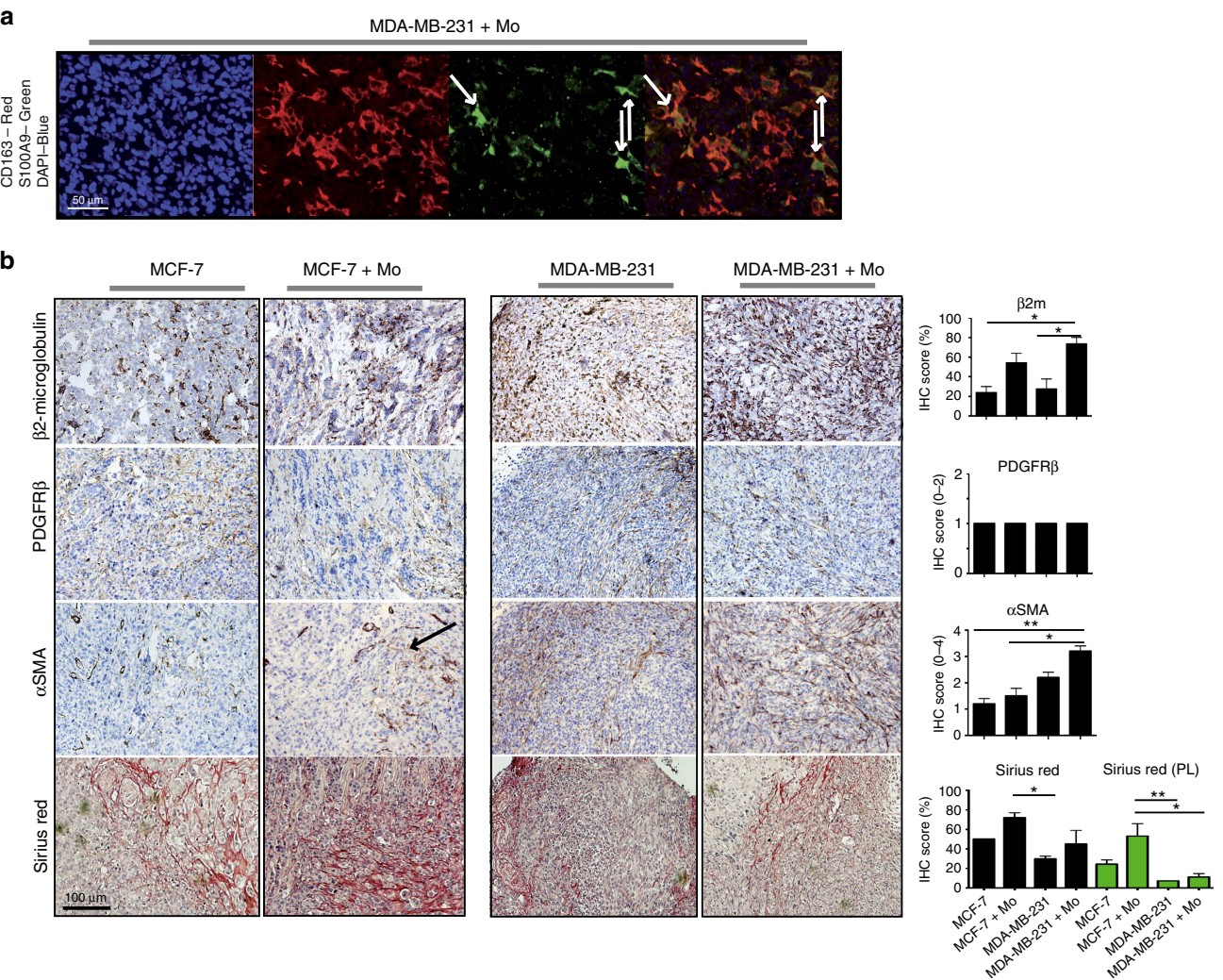

**Figure 2 | IHC of xenografts.** Tumour xenografts consisting of TN MDA-MB-231 breast cancer cells co-transplanted with primary human monocytes, express more activated stromal markers than luminal A MCF-7/monocyte xenografts. Xenografts were grown in highly immunodeficient NSG-mice (see 'Methods' section). (**a**) Immunofluorescence staining of CD163 (TRITC, red) and S100A9 (FITC, green) in sections from MDA-MB-231/monocytes xenograft tumours (day 21). DAPI (blue) shows nuclear staining. Overlay of colours is shown in lower right panel. S100A9 expression is located in the CD163[+] human myeloid co-transplanted cells. White arrows indicate three cells with co-expression of CD163[+] and S100A9. The cell lines are negative for S100A9 (ref. 21). (**b**) IHC was performed using the indicated antibodies and histological stains. All histological sections were counterstained with HE. $N = 5$ mice were analyzed for each group; The black arrows indicate activated stroma. MCF-7 grafts were analysed on day 21 and 90 post-graft with similar results—day 90 is shown here; TN MDA-MB-231 grafts were grown to day 21 only. The histograms show the mean value for each IHC score with statistical analyses. IHC scores are shown in Supplementary Table 1. $* = P < 0.05$, $** = P < 0.01$ ANOVA non-parametric Kruskal–Wallis test. $N = 5$. Error bars indicate s.e.m.

monocyte survival and proliferation in the TN grafts, we compared proteome arrays conducted on supernatants collected before and after co-culture with monocytes, using either TN (MDA-MB-231, MDA-MB-468 and SUM-159) or luminal A (MCF-7 and T47D) breast cancer cells (Supplementary Fig. 4C–E). The TNBC cell line supernatants collected before monocyte co-cultures showed a typical expression pattern of chemokines, angiogenesis and invasion related proteins (Supplementary Fig. 4C; purple box), whereas the luminal A breast cancer cells (blue box) expressed fewer factors. The proteins that were more upregulated in TNBC cell lines/ monocyte co-cultures than in luminal A/monocyte co-cultures were GM-CSF, MMP9, endothelin-1 and CXCL4 (platelet factor 4), and as shown previously, levels of IL-8 and CCL2 increased when monocytes were added to the TN breast cell lines and T47D cells[24] (Supplementary Fig. 4D,E; green and pink box). Thus, the TN tumour cell environment, alone or together with monocytes,

harbour important myeloid cell survival, proliferation and differentiation factors.

**Monocytes induce stroma formation.** To examine whether co-transplanted monocytes would affect stroma formation or activation, we next analysed the stromal component of the tumour xenografts. Recruited cells of mouse origin were present in all xenografts (mouse β2-microglobulin staining; Fig. 2), but the TN co-transplants had the highest number of recruited mouse cells (Supplementary Table 1). The recruited cells were of fibroblast origin since they expressed the fibroblast marker PDGFRβ (Fig. 2 and Supplementary Table 1).

As activated fibroblasts are central players in the tumour microenvironment we also analysed whether the stromal fibro-blasts were activated in the tumours using the marker αSMA (Supplementary Tables 1 and 2, Fig. 2 and Supplementary Fig. 1).

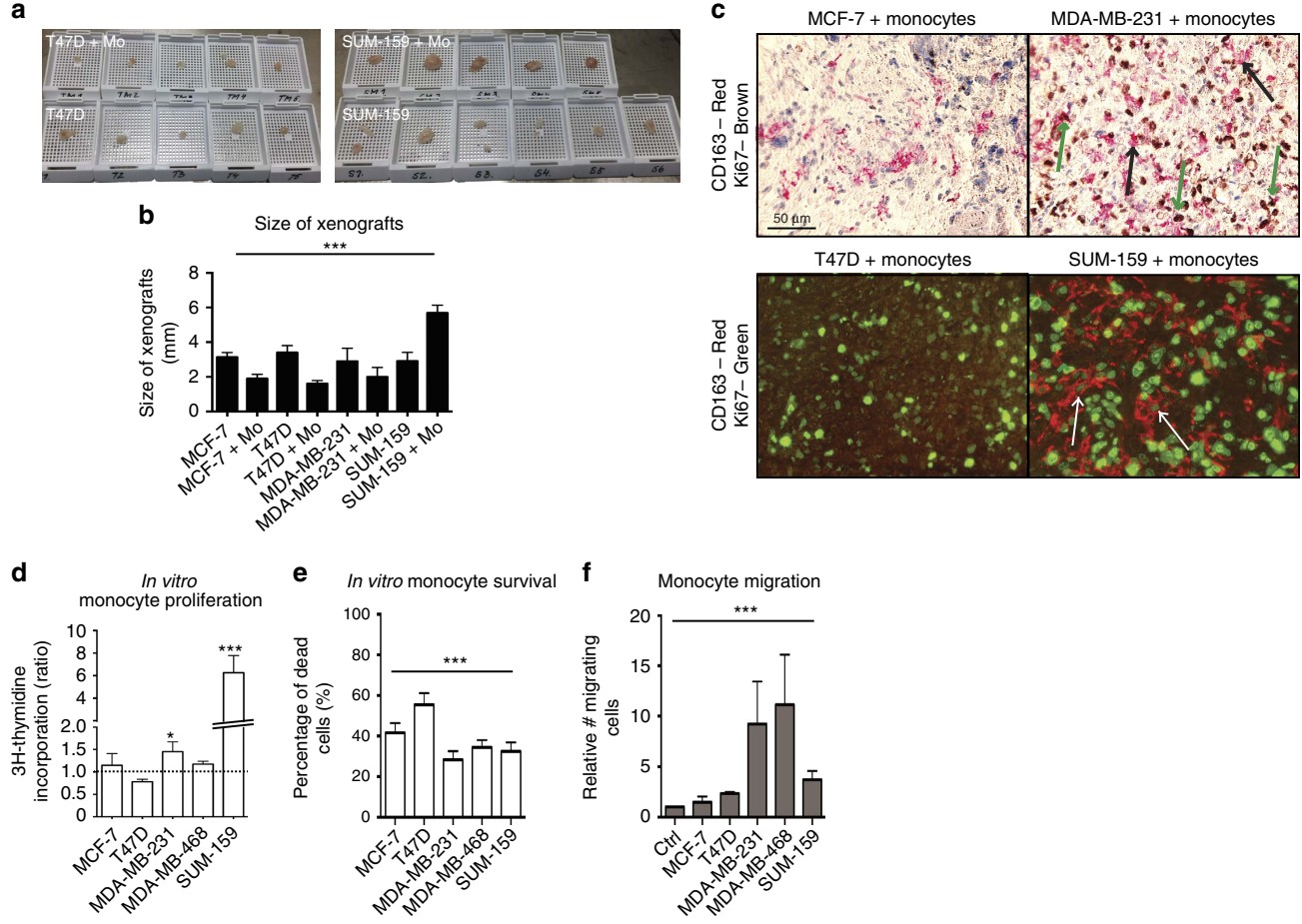

**Figure 3 | Characterization of primary monocytes in TNBC cultures.** Primary human monocytes show increased proliferation, survival and migration in a TNBC/monocyte environment than in a Luminal A/monocyte context. (**a,b**) The size of the xenografts differed between groups. Co-transplantation of monocytes generally decreased the tumour size slightly (**b**) except in the TN SUM-159/monocyte group, co-transplantation of monocytes (SUM-159 + Mo) increased tumour growth significantly (**a,b**). *** = $P < 0.001$ ANOVA. $N = 5$. (**c**) Myeloid cells proliferate *in vivo*. Upper panel: Double staining IHC of CD163 and Ki67 in xenografts from MCF-7/monocytes co-transplant (left) or MDA-MB-231/monocyte co-transplants (right) tumours as indicated. Black arrows show single staining with only CD163 and green arrows show double staining. Lower panel: Double staining immunofluorescence of CD163 and Ki67 in xenografts from T47D/monocytes co-transplant (left) or SUM-159/monocyte co-transplants (right) tumours as indicated. White arrows show CD163 + cells with a clear nuclear Ki67 staining. (**d**) Proliferation of primary human monocyte cultured in control medium or in conditioned medium from different cell lines, was assessed using the thymidine incorporation assay. * = $P < 0.05$ ANOVA. $N = 14$. (**e**) Survival of isolated human primary monocytes in breast cancer cell conditioned medium, grown for 7 days, was assessed. Annexin V staining was performed to analyse the total content of apoptotic/dead cells. *** = $P < 0.001$. ANOVA. $N = 5$. Error bars indicate s.e.m. (**f**) Boyden chamber migration assay of primary human monocytes migrating towards control medium or MCF-7, T47D, MDA-MB-231, MDA-MB-468 or SUM-159 breast cancer cell conditioned medium. *** = $P < 0.001$ ANOVA. $N = 5$. Error bars indicate s.e.m.

MCF-7 and T47D tumours that had been co-transplanted with monocytes showed a very modest, if any increase in fibroblast αSMA expression (Fig. 2, Supplementary Fig. 1 and Supplementary Tables 1 and 2). By contrast, the MDA-MB-231 and SUM-159 tumours that had been co-transplanted with monocytes showed a major increase in fibroblast αSMA expression (Fig. 2, Supplementary Fig. 1 and Supplementary Tables 1 and 2). These fibroblasts also showed the typical spindle shaped morphology in the TN reactive stroma (Fig. 2). Since the αSMA antibody recognizes both human and mouse αSMA, we co-stained with the human myeloid cell markers CD163 as well (Supplementary Fig. 5A). Most αSMA + cells were CD163 negative and therefore of mouse origin (black arrows), although a few of the αSMA + cells were of human origin and stained positive for CD163 (green arrows; Supplementary Fig. 5A). The αSMA + cells did not stain for the mouse myeloid marker Gr-1 (Supplementary Fig. 2C). Altogether, this indicates that the majority of αSMA + cells are activated mouse fibroblasts, and

that they are significantly increased in the TN xenografts co-transplanted with monocytes.

**The tumour stroma in TN PDX and syngeneic models.** To verify that our findings were applicable in other preclinical models, we next generated PDXs using tumour tissue from one luminal (HCI-011) and four TN (HCI-001, 002, 004 and 010) breast cancers (Fig. 4). We also generated syngeneic mouse tumours from one luminal (67NR) and one TN (4T1.13) breast cancer cell line (Supplementary Fig. 5B,C). As indicated in Fig. 4, there were significantly more myeloid cells of mouse origin (Ly6C; Fig. 4a,b), significantly more S100A9-expressing cells of mouse origin (Fig. 4a,b) and significantly more activated fibroblasts as indicated by the increased αSMA staining (Fig. 4a,b) in the TN PDX grafts as compared with the luminal PDX graft.

Similarly, Ly6C, mouse S100A9 and αSMA expression were increased in the TN as compared with the luminal grafts in the

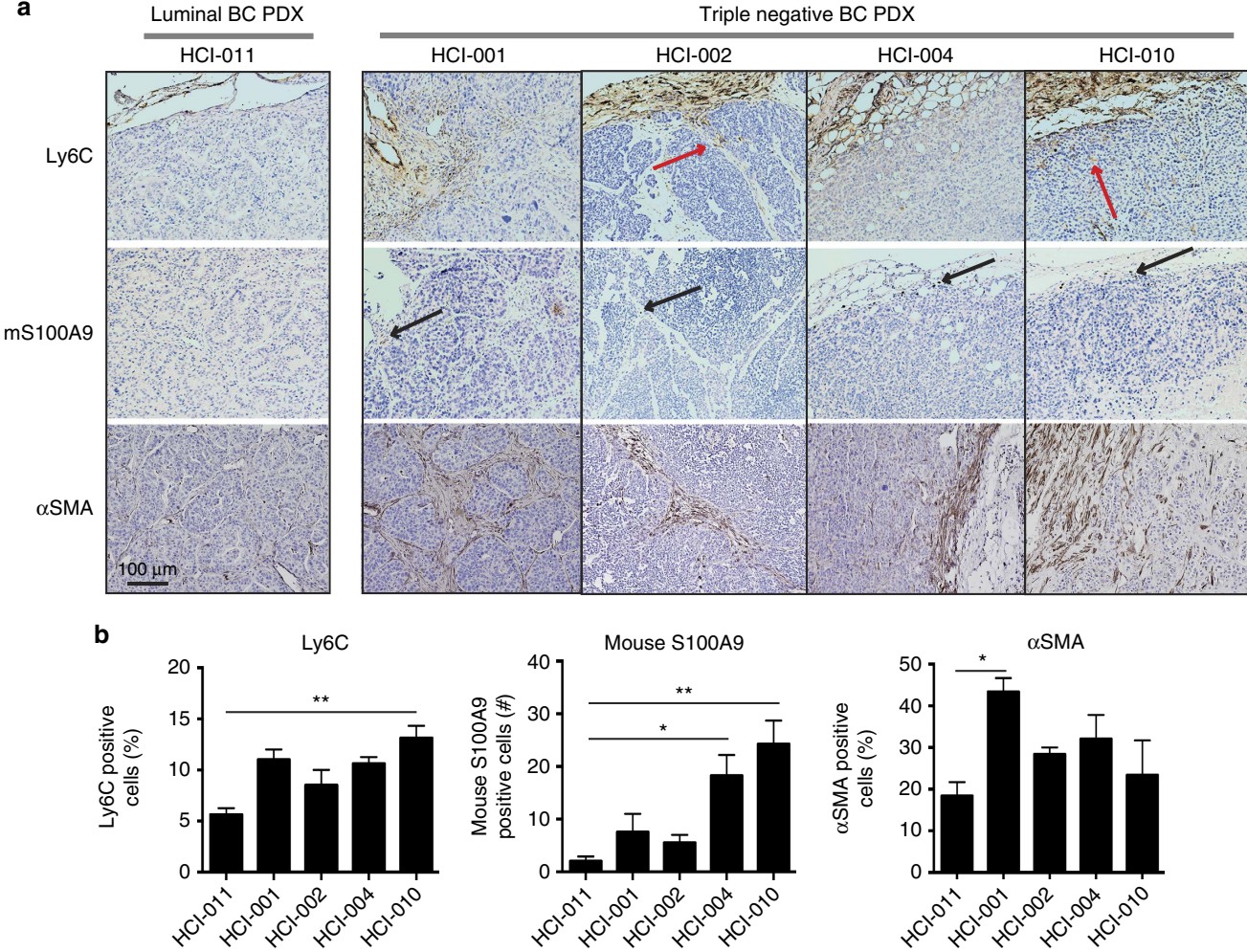

**Figure 4 | IHC of PDXs.** PDXs from TN breast tumours show increased myeloid cell infiltration, expression of S100A9 and activated fibroblasts. PDXs from one luminal (HCI-011) and four TN (HCI-001, HCI-002, HCI-004 and HCI-010) breast cancers grown in non-obese diabetic-severe combined immunodeficiency (NOD-SCID) mice were analysed for presence of monocytes (Ly6C mouse specific), expression of mouse S100A9 (mouse specific) and activated fibroblast (αSMA; recognizes human and mouse). (**a**) Ly6C positive cells were present in the tumour borders, with some cells infiltrating the stromal areas in particular (red arrows). Mouse S100A9 positive cells were present in the tumour borders and areas where myeloid cells were also present, and importantly mainly in the TN PDX grafts (black arrows). αSMA was expressed in the stromal areas of the PDX grafts, representing activated fibroblasts. (**b**) Quantitation of the immunohistochemical stains as presented by the histograms showing the mean value for each protein. For each PDX graft, 2–4 sections were stained and scored ($N = 3–6$). For αSMA and Ly6C the percentage (%) of positive cells was scored and for mouse S100A9 the numbers ($n$) of infiltrating cells expressing S100A9 was scored. A significantly higher level of Ly6C positive cells (left), of mouse S100A9 positive cells (center) and of αSMA (right) was seen in the TN PDX grafts as compared with the luminal PDX graft. $* = P < 0.05$ $*** = P < 0.001$. ANOVA non-parametric Kruskal–Wallis test. $N = (3–6)$. Error bars indicate s.e.m.

syngeneic model systems (Supplementary Fig. 5B,C). The increased expression of S100A9 in the TN grafts has previously been reported to associate with an increased metastatic property in the TN 4T1.2 as compared with luminal 67NR grafts[25]. Indeed, when mRNA expression profiles of the syngeneic grafts were analysed, also *ACTA2* (αSMA) was significantly increased in whole tumour all exon array data of 4T1.2 versus 67NR tumours (Supplementary Table 3)[25]. S100A9 mRNA was not upregulated *in vitro*, a finding that might indicate *in vivo* requirements for S100A9 expression or be explained by its post-transcriptional regulation[26].

**Less classical collagen depositions in TN xenografts.** Collagen is the major component of the ECM and the tumour microenvironment actively promotes degradation and re-deposition of collagen to promote tumour progression[27]. Collagens can be divided into fibrillar (for example, Type I, II, III, V) and non-fibrillar collagens (for example, Type IV and VI). Collagen

IV is a basement membrane collagen and collagen VI is a beaded filament-forming non-classical collagen that is associated with ECM[28]. The main collagens present in tumours are Type I, III, IV and VI (refs 27,29). Myeloid cells play an important role in the ECM remodelling by producing matrix metalloproteinases (MMPs) that degrade collagen[30].

To investigate the effects of co-transplanted myeloid cells on the collagen content in our xenografts, we used the Sirius Red stain (Supplementary Tables 1 and 2, Fig. 2, Supplementary Figs 1 and 6A,B). Under bright field microscopy (Fig. 2, Supplementary Figs 1 and 6A,B) Sirius Red detects collagens of type (I, III and IV) and using polarized birefringent light microscopy only collagens of type I and III are detected (red/green/yellowish staining; Supplementary Fig. 6A,B)[31]. We found that, co-transplantation of monocytes slightly increased the collagen deposition in all of the grafts (Supplementary Tables 1 and 2; Fig. 2, Supplementary Figs 1 and 6A,B). The MCF-7/monocyte co-transplants on day 21 and day 90 showed robust collagen

deposition (Supplementary Table 1; red colour, Fig. 2 (day 90); red colour and green/yellow, Supplementary Fig. 6A,B (day 21 and 90)). The T47D/monocyte co-transplants showed a similar collagen deposition (Supplementary Table 2 and Supplementary Fig. 1), whereas the TN MDA-MB-231/monocyte and SUM-159/monocyte co-transplants showed a larger variation between tumours (25–75% stroma Supplementary Tables 1 and 2; red colour, Fig. 2, Supplementary Figs 1 and 2A). Interestingly, despite the high number of fibroblasts in the TN MDA-MB-231/monocyte and SUM-159/monocyte xenografts (Fig. 2 and Supplementary Fig. 1), they expressed significantly less collagens of type I and III, compared with the MCF-7/monocyte and T47D/monocyte co-transplants (Supplementary Tables 1 and 2; red/green/yellowish pictures; Supplementary Fig. 6A,B). A low expression of collagens type I and III in TNBCs, specifically, was also verified in a tissue microarray consisting of 144 human breast cancers, where a low birefringent light of Sirius Red viewed in polarized light correlated significantly to TN breast tumours (Table 1; $P = 0.036$). Bright light Sirius Red staining of collagen deposition did not correlate to TNBCs, but showed a negative correlation to both tumour size ($P = 0.003$) and Nottingham histologic grade (NHG) status ($P = 0.002$) (Table 1); Spearman's Rho analysis using SPSS software.

The non-classical beaded filament collagen VI was expressed primarily in the TN MDA-MB-231/monocyte and SUM-159/monocyte xenografts, but also to some extent in the SUM-159 xenografts (Supplementary Tables 1 and 2; Supplementary Fig. 7A)[28]. The collagen VI deposits in the MDA-MB-231/monocyte xenografts probably came from the transplanted myeloid cells, since primary human monocytes cultured in conditioned medium from TN MDA-MB-231 and MDA-MB-468, but not MCF-7 and interestingly not SUM-159 breast cancer

cells, upregulate collagen VI to a similar level as the control M2 macrophages (Supplementary Fig. 7B,C)[28]. Primary mouse fibroblasts cultured in conditioned medium from breast cancer cells, or from co-cultures of breast cancer cells and primary monocytes, did not express more collagen VI than mouse fibroblasts grown in normal medium (Supplementary Fig. 7B,C). Monocytes cultured in conditioned medium from the luminal T47D breast cancer cells also induced collagen VI, indicating that it might be a more general breast cancer cell inducing mechanism, unrelated to the TNBC subtype (Supplementary Fig. 7B,C).

These findings suggest that the tumour type will direct the myeloid cells differently in TN as compared with luminal A tumours, so that myeloid cells will produce both collagen degrading MMPs and perhaps also anti-inflammatory collagen VI in TN tumours, thus promoting their invasiveness[28]. Thus, although the TN MDA-MB-231/monocyte and SUM-159/monocyte co-transplants had more fibroblasts as judged by IHC (Supplementary Tables 1 and 2, Fig. 2 and Supplementary Fig. 1) these fibroblasts did not produce more of the classical collagens.

**Primary fibroblasts are activated by monocytes in TN tumours.** We next investigated what might facilitate the high number of mouse fibroblasts seen in the TN MDA-MB-231/monocyte and SUM-159/monocyte co-transplants. To this end, we performed primary mouse fibroblast migration, survival and proliferation assays *in vitro*, to compare the effect of TN/monocyte and luminal/monocyte conditioned medium.

Scratch wound assays revealed that primary mouse fibroblasts migrated equally well in supernatants from all culture conditions (Fig. 5a and Supplementary Fig. 7D), while a significant survival advantage was seen only in the primary mouse fibroblasts

**Table 1 | Analysis[a] of CXCL16 and Collagen expression in a breast cancer tissue microarray.**

|  | Sirius Red | Low birefringent Sirius Red (PL) | CXCL16 malignant cells | CXCL16 fibroblasts |
|---|---|---|---|---|
| *TNBC* |  |  |  |  |
| Correlation coefficient | − 0.142 | 0.178* | − 0.025 | 0.231** |
| Sig. (2-tailed) | 0.104 | 0.036 | 0.779 | 0.010 |
| N | 132 | 139 | 129 | 123 |
| *Tumour size* |  |  |  |  |
| Correlation coefficient | − 0.263** | 0.130 | − 0.099 | − 0.088 |
| Sig. (2-tailed) | 0.003 | 0.122 | 0.256 | 0.326 |
| N | 136 | 144 | 133 | 126 |
| *Node status* |  |  |  |  |
| Correlation coefficient | − 0.028 | − 0.093 | − 0.182* | − 0.057 |
| Sig. (2-tailed) | 0.759 | 0.294 | 0.048 | 0.546 |
| N | 122 | 129 | 119 | 115 |
| *NHG* |  |  |  |  |
| Correlation coefficient | − 0.257** | 0.031 | − 0.137 | − 0.098 |
| Sig. (2-tailed) | 0.002 | 0.715 | 0.117 | 0.277 |
| N | 136 | 144 | 133 | 126 |
| *Sirius Red* |  |  |  |  |
| Correlation coefficient | — | − 0.291** | − 0.035 | 0.032 |
| Sig. (2-tailed) |  | 0.001 | 0.692 | 0.723 |
| N |  | 136 | 132 | 125 |
| *Low birefringent Sirius Red (PL)* |  |  |  |  |
| Correlation coefficient | − 0.291** | — | 0.283** | 0.220* |
| Sig. (2-tailed) | 0.001 |  | 0.001 | 0.013 |
| N | 136 |  | 133 | 126 |

[a]Spearman's Rho analysis using SPSS software. *$P < 0.05$; **$P < 0.01$; ***$P < 0.001$.
Correlation between expression levels and clinical parameters in a breast cancer tissue microarray ($N = 144$) of CXCL16 in malignant cells (0–3), CXCL16 in fibroblasts (0–1), Sirius Red (0–3) or Low birefringent in polarized light (PL) Sirius Red (Birefringent in PL (0)—No birefringent in PL (1)). For scoring see 'Methods'.

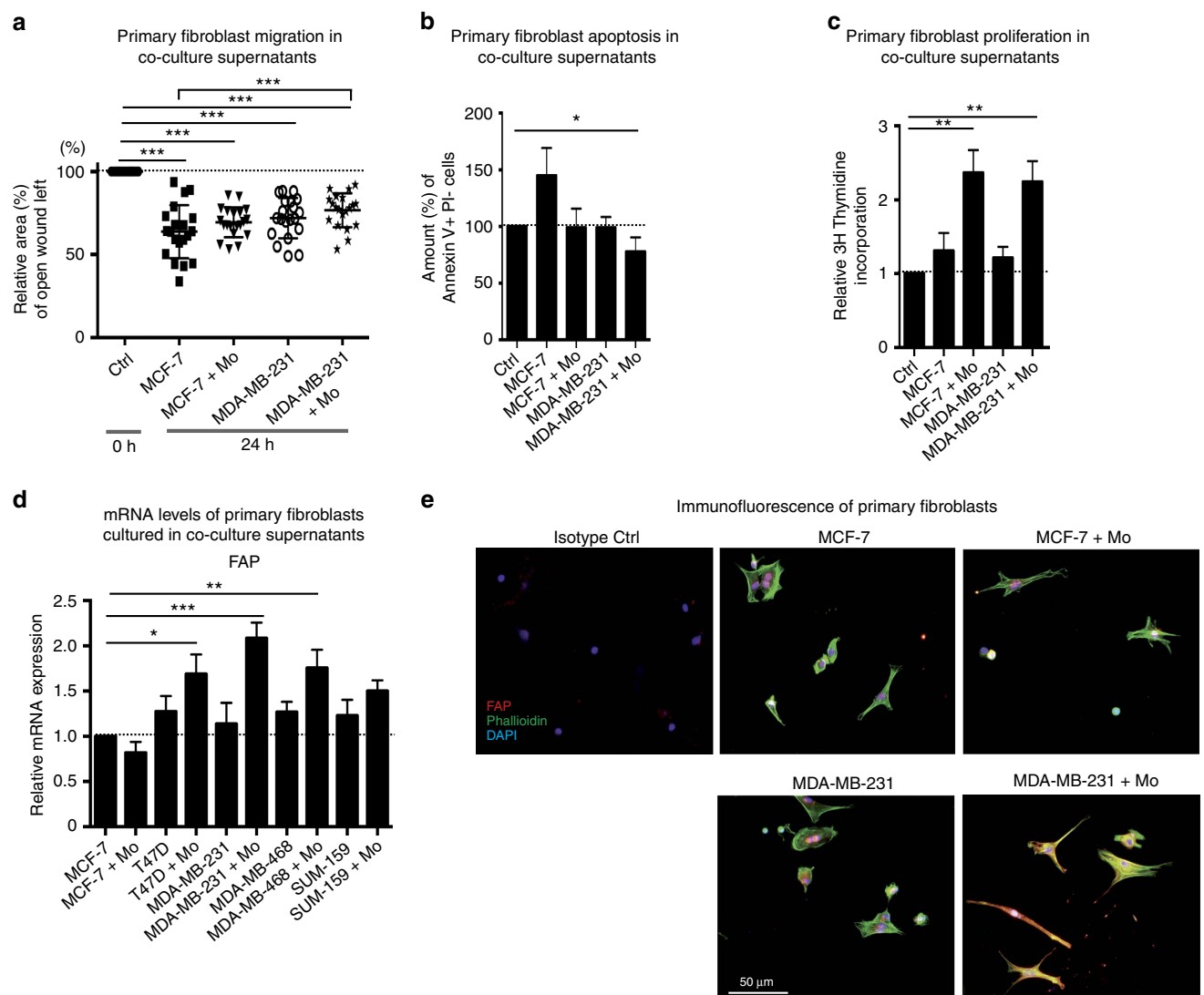

**Figure 5 | Characterization of primary fibroblasts in TNBC cultures.** Primary mouse fibroblasts are activated by monocytes in the TNBC context *in vitro*. (**a**) Scratch wound assays showing mouse primary fibroblast migration in supernatants derived from co-cultures of human primary monocytes (Mo) and luminal A (MCF-7) or TN (MDA-MB-231) breast cancer cells. *** = P < 0.001 ANOVA non-parametric Kruskal–Wallis test. N = 20. (**b**) Survival analysis of mouse primary fibroblast grown in supernatants derived from co-cultures of human primary monocytes and luminal A (MCF-7) or TN (MDA-MB-231) breast cancer cells. Annexin V staining was used to analyse the percentage of apoptotic cells. * = P < 0.05 ANOVA. N = 10. (**c**) Proliferation of mouse primary fibroblasts grown in supernatants derived from co-cultures of human primary monocytes and luminal A (MCF-7) or TN (MDA-MB-231) breast cancer cells, measured using a thymidine incorporation proliferation assay. ** = P < 0.01. ANOVA Dunnett's multiple comparison test. N = 14. (**d**) mRNA expression levels of FAP in mouse primary fibroblasts cultured in supernatants derived from co-cultures of human primary monocytes and luminal A (MCF-7 and T47D) or TN (MDA-MB-231, MDA-MB-468 and SUM-159) breast cancer cells, assessed by RT-QPCR analysis. * = P < 0.05 ** = P < 0.01 *** = P < 0.001 ANOVA Dunnett's multiple comparison test. N = 4. (**e**) Immunofluorescence of anti-fibroblast activation protein (FAP; red), phalloidin to stain actin filaments (green) and DAPI (nuclear stain; blue) in primary mouse fibroblasts cultured in supernatants derived from co-cultures of primary human monocytes and luminal A (MCF-7) or TN (MDA-MB-231) breast cancer cells. Scale bar represents 50 μm. Error bars indicate s.e.m.

cultured in TN MDA-MB-231/monocyte and SUM-159/monocyte supernatant (Fig. 5b and Supplementary Fig. 7E). Increased proliferation of mouse fibroblasts was seen in MCF-7/monocyte, MDA-MB-231/monocyte and SUM-159 or SUM-159/monocyte cultures (Fig. 5c and Supplementary Fig. 7F), but not in T47D/monocyte or MDA-MB-468/monocyte co-cultures (Supplementary Fig. 7F). These data indicate that fibroblast survival is probably the major cause of the increased presence of fibroblasts in the TN MDA-MB-231/monocyte and SUM-159/monocyte grafts, but that fibroblast proliferation also can be affected significantly by myeloid cells.

The activation status of cultured primary mouse fibroblasts was then investigated by measuring fibroblast activating protein

(FAP) levels. Interestingly, we found that primary mouse fibroblasts were activated by all of the TNBC cell/monocyte supernatants, as seen by upregulation of FAP at both mRNA (Fig. 5d) and protein (Fig. 5e) levels, further corroborating our αSMA data presented in Fig. 2 and Supplementary Fig. 1. The luminal T47D/monocyte supernatants also upregulated fibroblast FAP mRNA (Fig. 5d), while fibroblast TGFβ mRNA was induced slightly only by culture in the MDA-MB-231 conditioned medium (Supplementary Fig. 7G). The low numbers of myeloid cells surviving in the luminal T47D xenografts may explain why fibroblast activation was affected only in the *in vitro* cultures of T47D/monocytes supernatants, but not in the corresponding tumours.

**Fibroblasts in TN tumours produce CXCL16 specifically**. In the experiments mentioned above, we showed that monocytes can promote stroma formation equally well in TN and luminal A tumours, but that the activated fibroblasts are numerous in the TN tumours. This does not explain why CD163[+] myeloid cells are more frequent in the stromal areas of TN breast tumours, as we published previously[4]. We therefore looked for myeloid cell chemoattractants that would be produced specifically in fibroblasts from TN breast tumours. We first isolated primary human CAFs from ER[−] (TN) and ER[+] breast tumour patient samples, cultured them *in vitro* and collected the supernatants. We subsequently performed an angiogenesis protein array and found that CXCL16, amphiregulin and tissue inhibitor of metalloproteinases (TIMP1) were expressed at high levels in the TN but not the ER[+] breast cancer CAF supernatants (red box; Fig. 6a). CXCL16 is known to be a T cell chemoattractant, but recently also as a myeloid cell chemoattractant. Amphiregulin is a protein involved in tumour progression and tissue inhibitor of metalloproteinases act to inhibit MMPs[32–36]. IHC of CXCL16 on primary breast tumours revealed that both ER[+] and TN tumour cells expressed CXCL16 (black arrow; Fig. 6b), and that fibroblasts in the TN tumour also expressed CXCL16 (red arrow; Fig. 6b).

We next evaluated the levels of secreted CXCL16 using an enzyme-linked immunosorbent assay (ELISA) on supernatants from primary CAFs and showed that CAFs isolated from TN tumours (3/4) secreted high amounts of CXCL16, while low CXCL16 was measured in CAFs isolated from ER[+] tumours (0/8) (Fig. 6c). IHC staining of CXCL16 in a tumour tissue microarray consisting of 144 human breast cancers showed that high fibroblast expression of CXCL16, correlated significantly to TN breast tumours ($P = 0.010$), while CXCL16 expression in the malignant cells *per se* did not (Table 1). However, only a fraction of the TN tumours expressed CXCL16 in the fibroblasts (Table 2). The high fibroblast CXCL16 expression also correlated to a low birefringent light of Sirius Red ($P = 0.013$; Table 1). Boyden chamber migration experiments showed that monocytes (Fig. 6d left) and, to a lesser extent, M2 macrophages (Fig. 6d, right) migrated towards CXCL16 more so than towards another chemokine, CXCL12. We also observed a significant monocyte migration towards ER[−] (TN) CAF supernatants (Fig. 6e). The hematoxylin eosin (HE) image (Fig. 6e, right) shows a cytospin of the migrated cells. In support of these findings, we also found that expression of *CXCL16* mRNA was induced in activated primary mouse fibroblasts cultured in TN MDA-MB-231/monocyte or MDA-MB-468/monocyte conditioned medium, but not in conditioned medium from MDA-MB-231 or MDA-MB-468 cell culture without monocytes, nor the luminal cell supernatants (Fig. 6f). The TN SUM-159/monocyte supernatant also increased *CXCL16* mRNA expression significantly, as did the SUM-159 only supernatants (Fig. 6f). Using PDX models, mouse *CXCL16* mRNA originating from infiltrating mouse cells was significantly induced in one of the TN PDX grafts, and increased in the other three TN PDX grafts, as compared with the luminal PDX graft (Fig. 6g). Finally, in the syngeneic models *CXCL16* mRNA was significantly increased in whole tumour all exon array data of 4T1.2 versus 67NR tumours (Supplementary Table 3)[25]. Since CXCL16 can attract T cells, NKT cells, myeloid cells and also fibroblast precursors, these findings are of large importance when it comes to understanding why the tumour stroma in TN breast tumours in particular, attracts immune cells.

**Immunosuppressive gene expression profile in TNBCs**. To extend our findings from our preclinical models to primary human breast tumours, we analysed RNAseq data from The Cancer Genome Atlas. The tumours were sub-grouped based on the PAM50 centroids and here the basal subgroup largely coincides with TN tumours. While mRNA encoding the myeloid cell marker CD11b (*ITGAM*) was expressed at equal levels in all breast tumour subgroups (Fig. 7a), mRNA of the anti-inflammatory markers *CD163* and *S100A9* were expressed at significantly higher levels in basal than luminal A breast tumours (Fig. 7a). Also, the collagen VI (*COL6A1*)/collagen I (*COL1A1*) mRNA ratio (*COL6A1/COL1A1*) was significantly higher in basal than luminal A breast tumours (Fig. 7a). Further corroborating our findings, we found that mRNAs for *CXCL16* as well as for *IL8, CSF2, MMP9, EDN1* and *CCL2* all were expressed at significantly higher levels in basal-like breast cancers as compared with luminal A tumours (Fig. 7a).

**Discussion**

In this study, we demonstrate that co-transplanted primary human monocytes differentiate into CD163[+] myeloid cells, both in luminal A and TN breast tumour grafts, but that the immunosuppressive MDSC-marker S100A9 (refs 37,38) was expressed only by the CD163[+] cells in the TN grafts. S100A9 expression has previously been associated with ER[−]PR[−] tumours, both when expressed in the myeloid cells but also in the malignant cells *per se*[21]. Neither MDA-MB-231 nor SUM-159 cells express S100A9, thus making it possible for us to investigate the myeloid S100A9 expression, in a TN environment *in vivo*. More myeloid cells were present in the TN tumours and this was due to their increased survival and proliferation. Hence, the TN tumour environment is crucial for myeloid cells and their effect on fibroblasts. We also showed that monocytes are preferentially attracted to TN breast tumours by secreted factors, such as GM-CSF, CCL2 and IL-8, confirming the findings of Hollmén *et al.*[39] and Su *et al.*[40]. In addition, S100A9 is probably one of the major MDSC chemoattractants[16]. Even though expression of the mouse myeloid differentiation antigen Gr-1 was scarce, a potential infiltration of immature BMDCs cannot be excluded as NSG-mice still have normal numbers of immature myeloid cells[22]. Furthermore, the xenograft models used in this study represent a situation where the myeloid cells are present in the tumours from initiation and hence, these models do not address the question how these cells are recruited to the tumours or which cells of the microenvironment that arrive first. Therefore, we confirmed our data in mice grafted with TN as compared with luminal patient-derived tumour tissue (PDX), where we could see that more myeloid cells infiltrated the TN tumours, and although the infiltrating myeloid cells were located closer to the borders of the tumours, these areas did show more S100A9 expression and activated fibroblasts (αSMA) in the TN, as compared with the luminal graft. We observed a similar pattern using syngeneic mouse models of luminal (67NR) as compared with TN (4T1) breast tumours, grafted in immunocompetent mice. The effect of myeloid cells on tumour stroma formation was seen in the borders of grafts in general, and therefore might be underestimated in human tumour pathology. In our model where the myeloid cells were mixed with the tumour cells prior to injection, a clear effect was seen throughout the tumour, on both activated fibroblasts and myeloid cells *per se*. In the future, we suggest validating our findings in spontaneous breast cancer models with transplanted circulating human myeloid cells, but also in models using novel drugs against tumour-infiltrating myeloid cells.

Monocytes have been previously shown to promote metastasis but evidence is lacking on how they affect the tumour stroma[18]. We showed that the myeloid cells in both luminal A and TN tumours enhanced recruitment of fibroblasts of mouse origin, but

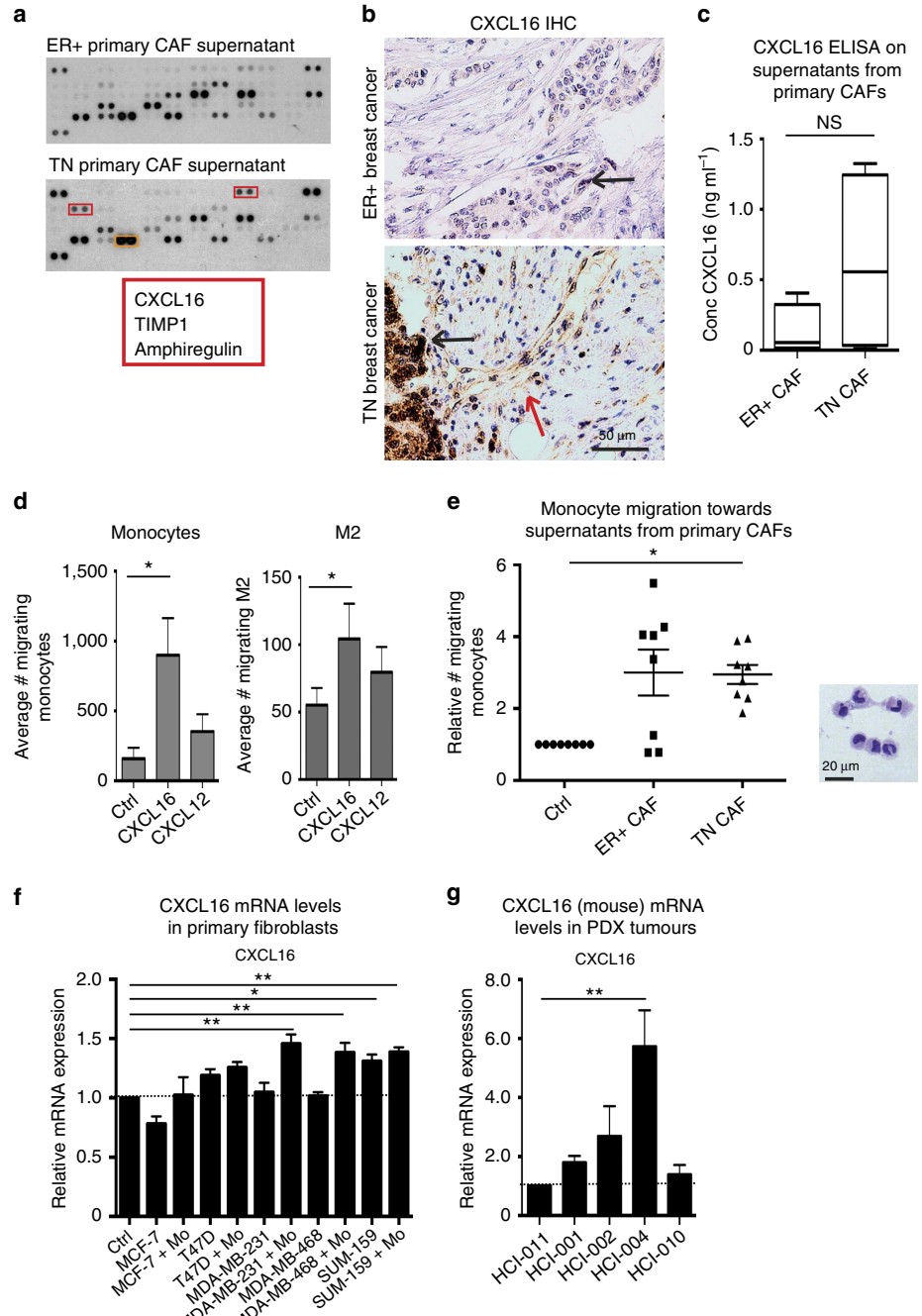

**Figure 6 | CXCL16 expression is induced in fibroblasts by myeloid cells.** (**a**) Human angiogenesis array proteome profiler of supernatants from primary CAFs derived from ER$^+$ or ER$^-$PR$^-$Her2$^-$ (triple negative; TN) tumours. (**b**) IHC of CXCL16 in human primary breast cancers. Upper panel shows ER$^+$PR$^+$ breast cancer and lower panel one TNBC. Black arrow highlights CXCL16 expressing malignant cells and red arrow, CXCL16 expressing fibroblasts. (**c**) The levels of secreted CXCL16 were measured using a human CXCL16 ELISA performed on supernatants prepared from primary human CAFs isolated from TN or ER$^+$ primary breast tumours. $N = 4$ TN and $N = 8$ ER$^+$. Mann–Whitney $U$-test. (**d**) Migration of primary human monocytes (left) or M2 macrophages (right) was measured using Boyden migration chambers with 8 μm pore size towards control medium (Ctrl), recombinant CXCL16 or CXCL12 as a control. $* = P < 0.05$. ANOVA Dunn's multiple comparisons test. $N = 5$ and $N = 14$. (**e**) Migration of primary human monocytes (left) or M2 macrophages (right) was measured using Boyden migration chambers with 8 μm pore size towards control medium (Ctrl), or towards conditioned medium from primary human CAFs derived from either ER$^+$ or TN tumours. $* = P < 0.05$ $** = P < 0.01$. ANOVA Dunn's multiple comparisons test. $N = 8$. The HE image (right) shows that the migrated cells are indeed monocytes and the scale bar represents 20 μm. (**f**) RT-QPCR analysis of CXCL16 levels in primary mouse fibroblasts cultured in either control medium (Ctrl) or supernatants derived from luminal A (MCF-7 and T47D) or TN (MDA-MB-231, MDA-MB-468 and SUM-159) breast cancer cells with or without co-culture with primary human monocytes (Mo). $* = P < 0.05$ $** = P < 0.01$. ANOVA Dunn's multiple comparisons test. $N = 4$–$8$. Error bars indicate s.e.m. (**g**) RT-QPCR analysis of infiltrating mouse cell CXCL16 levels in human PDX tumours derived from one luminal (HCI-011; set to 1) and four TN (HCI-001, 002, 004, 010) breast tumours. $** = P < 0.01$. ANOVA Dunn's multiple comparisons test. $N = 6$–$7$. Error bars indicate s.e.m.

**Table 2 | Crosstable[a] over TNBCs and CXCL16 fibroblast expression.**

|  | CXCL16 fibroblasts (0) | CXCL16 fibroblasts (1) | Total |
|---|---|---|---|
| TNBC 0 | 101 | 7 | 108 |
| 1 | 11 | 4*[a] | 15 |
| Total | 112 | 11 | 123 |

[a]$\chi^2$ Linear by linear Association $P = 0.011$ using SPSS software. *$P < 0.05$.

promoted their activation into myofibroblasts only in the context of TN tumours. Monocytes *per se* had the potential to activate the fibroblasts in the TN tumours and to induce an alternative collagen (collagen VI) formation. Hence, monocytes can promote stroma formation in both luminal A and TNBCs, but the type of stroma potentiated by the myeloid cells might depend on the tumour type. It is intriguing to note that in 3 out of 4 xenograft-groups, the monocyte co-transplanted tumours were smaller in size, a finding that might be explained by a different ECM in these tumours and the levels of MMPs (for example, MMP9) that can dictate the thickness of collagen deposition and also the orientation of collagen fibres[41]. Collagen VI, which is induced by TGFβ in macrophages, is not preferentially degraded by MMPs. It has also been shown to be a fibroblast mitogen and suggested to be involved in drug resistance, tumour progression and myeloid cell recruitment[29,42,43]. The beaded microfilament structure of collagen VI is also important for anchoring cells to the ECM and may be important for anchoring myeloid cells[28]. Undoubtedly, more research on the role for different collagens in tumour progression is warranted.

CXCL16 has previously been shown to attract T cells, bone marrow-derived fibroblast precursors and to potentiate fibrosis and myofibroblast activation in renal fibrosis[34,35,44]. In this study we show that monocyte-induced fibroblast activation involves expression of the chemoattractant CXCL16 in a TN context primarily. This observation is supported by our findings that primary human CAFs from TN breast tumours specifically expressed CXCL16 and that a high CXCL16 expression in fibroblasts correlates with a TN breast tumour type in a breast cancer tissue microarray. The CXCL16 expression levels varied, and not all TN breast tumours expressed CXCL16 in their fibroblasts, an observation that might be explained by the heterogeneity within the TN subgroup and indeed when analysed using the TCGA breast cancer RNAseq data, only the basal breast cancer subgroup showed a significantly higher expression level of *CXCL16*. We also found that CXCL16 could be expressed by the malignant cells of both luminal A and TN tumours, a finding that might explain why CD163[+]/GRN[+] myeloid cells are located in the tumour nests of luminal A tumours[4]. In light of this, we suggest that CXCL16 may be viewed as a monocyte and fibroblast chemoattractant expressed in human breast tumours. In addition to this, also GM-CSF (ref. 45) and CXCL4 (refs 46,47) are important for fibroblast recruitment or activation. Similarly, Su *et al.* and Hollmén *et al.* reported on a positive feedback loop between breast cancer cells and macrophages in mesenchymal-like and TNBCs[39,40].

In summary, the findings from our study indicate that myeloid cells are recruited preferentially to TN breast tumours where they become skewed to immunosuppressive myeloid cells (CD163[+] S100A9[+]), activate CAFs and induce expression of CXCL16 in CAFs that in turn can recruit more myeloid cells and fibroblasts, but also T cells (Fig. 7b). This is the first study to address the particular effects of myeloid cells on tumour stroma, and also specifically with regards to breast cancer subtype. These findings are of particular importance for the design of trials concerning

novel drugs being developed against tumour-infiltrating myeloid cells. In addition, our data add valuable information to aid management decisions concerning which drugs to be used in different breast cancer subtypes and also to predict drug resistance since anti-PD1/L1 resistant tumours might develop immune resistance in the presence of infiltrating anti-inflammatory myeloid cells. Our data suggest that patients with TNBCs, in particular, might benefit from treatment with novel immunomodulatory drugs or chemotherapeutics that target immunosuppressive cells.

## Methods

**Ethics statement.** Permission for the study was obtained from the Regional Ethics Committee at Lund University. For permission to conduct human research, the following ethics permits were obtained: Dnr 2010/477, Dnr 2012/689, Dnr 2014/669, Dnr 445/07 and Dnr 2009/658. The participating patients provided written informed consent or for the tumour tissue microarray had the option to withdraw. The NSG models (approvals M249-09 and M69/11) and nude mice (approval M149/14), were approved by regional ethics committee for animal research at Lund University, Sweden. The PDX experiments were reviewed and approved by the University of California, San Francisco Institutional Animal Care and Use Committee (IACUC). The syngeneic, mouse breast cancer models, were approved by the Animal Experimentation and Ethics Committee (AEEC) of the Peter MacCallum Cancer Center, Australia.

**Isolation of primary human monocytes.** Leucocytes were isolated within 2–3 h of blood collection from healthy blood donors by leucocyte depletion filtration performed according to a previously published method[48] or from blood collected in EDTA tubes from healthy blood donors. First, peripheral blood mononuclear cells were prepared using Ficoll-Paque PLUS gradient centrifugation, then monocytes were isolated by magnetic cell sorting using the Monocyte Isolation Kit II, according to the manufacturers' instructions (Miltenyi Biotec, Bergisch Gladbach, Germany) with antibodies against CD3, CD7, CD16, CD19, CD56, CD123 and glycophorin A.

**Animal procedures.** Female 8-week-old NSG-mice (NOD.Cg-*Prkdc(sci-d)Il2rg(tm1Wji)*/SzJ strain, The Jackson Laboratory, Maine, USA) were housed in a controlled environment and all procedures were approved by the regional ethics committee for animal research at Lund University, Sweden (approvals M249-09 and M69/11). MCF-7, MDA-MB-231, T47D or SUM-159 human breast cancer cell lines ($1 \times 10^6$ cells, $5 \times 10^6$ for T47D cells) were injected alone or in combination with primary human monocytes ($1 \times 10^6$ cells) that were pre-stimulated with interleukin-10 (IL-10) 10 ng ml$^{-1}$ for 30 min in a total volume of 100 µl Hanks' balanced salt solution to enhance survival[49], on the right flank. Mice were monitored twice weekly. Tumours were excised either 21 days (MDA-MB-231, T47D, SUM-159 and MCF-7) or 21 and 90 days (MCF-7) after injection, before being fixed in 4% paraformaldehyde and embedded in paraffin. For each experiment, 5–10 mice were used in each group. Similar results were obtained with MCF-7 grafts on day 21 and 90; results for day 90 are shown in Figs 1 and 2 and for day 21 in Supplementary Fig. 6.

NMRI-Nude mice (8-week-old) (The Jackson Laboratory Maine, USA) were housed in a controlled environment and all procedures were approved by the regional ethics committee for animal research (approval M149/14). Primary mouse fibroblasts were isolated by dissecting the ears of nude mice, cutting them into small pieces and treating overnight with collagenase type I (17100-017, Thermo Scientific, MA, USA) together with Dulbecco's modified Eagle medium (DMEM) high glucose supplemented with penicillin/streptomycin and 1% minimum essential medium Eagle (MEM) at 37 °C. The following day a single cell suspension was prepared, filtered through a 70 µm pore filter, washed and cultured in HAMs F12 medium supplemented with penicillin/streptomycin and 10% FBS. Subsequently, non-adherent cells were washed away.

**Patient-derived xenografts.** Breast cancer PDXs were generated from one luminal (HCI-011) and four TN (HCI-001, HCI-002, HCI-004, HCI- 010) breast cancers serially passaged in NOD/SCID mice as described previously[50,51]. Tumour fragments were placed into cleared inguinal fat pads of pre-pubescent NOD/SCID mice, grown to 20–25 mm and subsequently dissected and stored by freezing in 90% FBS and 10% dimethylsulfoxide until used or fixed in paraffin for IHC. Total RNA was extracted according to the manufacturers' instructions using RNeasy Plus Mini kit (Qiagen, Hilden, MD, USA) and iScript Reverse Transcription Supermix for RT-QPCR (BIO-RAD) was used for cDNA synthesis. RT-QPCR was performed using iTAG Universal SYBR Green Supermix (BIO-RAD).

**Syngeneic mouse breast cancer models.** Syngeneic primary tumours that were generated previously by injection of $1 \times 10^5$ luminal (67NR) or TN (4T1.13) cells in the fourth inguinal fat pad of 8–10-week-old female BALB/c mice[25] were used for IHC.

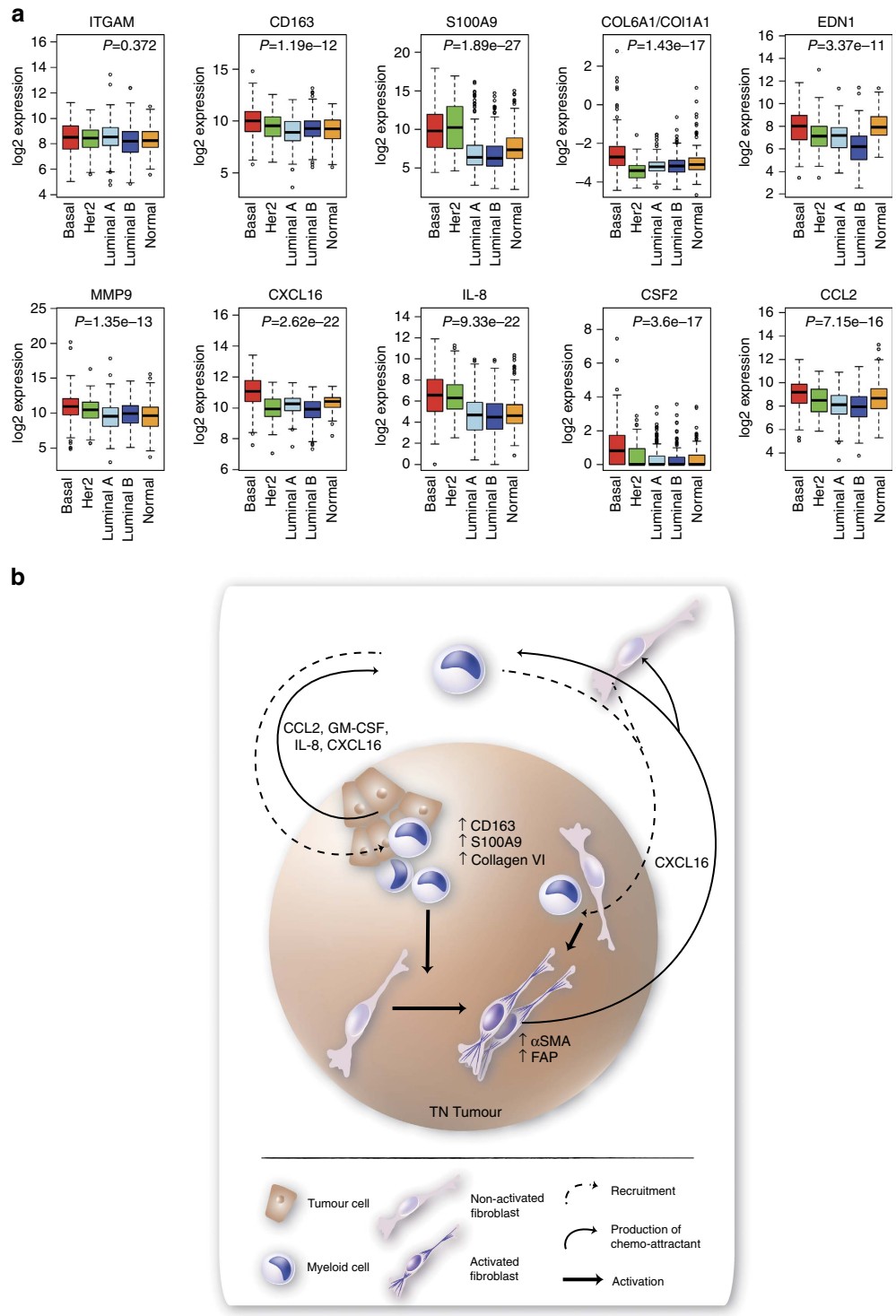

**Figure 7 | Gene expression profile in human breast cancers.** Human basal breast cancers have an immunosuppressive gene expression profile. (**a**) Box plots showing log2 gene expression levels of indicated genes in molecular breast cancer subtypes, using TCGA breast cancer RNAseq data. *P* values were calculated using a *t*-test comparing levels between basal-like tumours and luminal A tumours. The middle line demonstrates the median, the box illustrates the interquartile range, and the whiskers indicate the most extreme data point that is not >1.5 × the interquartile range away from the box. Data points beyond these values are individually shown. (**b**) Schematic to model the effects exerted by myeloid cells on stroma formation in TN breast tumours. Myeloid cells are recruited to TN breast tumours by proteins including CCL2, GM-CSF, IL-8, S100A9 and CXCL16, where they are induced to express CD163 and the immunosuppressive factors S100A9 and collagen VI. Furthermore, in a TN environment, the myeloid cells activate CAFs and induce expression of CXCL16 by the fibroblasts, which, in turn, can recruit more myeloid cells and fibroblasts. The activated stroma in combination with the presence of anti-inflammatory myeloid cells, will render the TN tumours a more aggressive behaviour.

**Cell culture.** The human breast cancer cell lines MDA-MB-231, MDA-MB-468, MCF-7 and T47D were obtained from ATCC, and cultured in 10% FBS RPMI-1640 supplemented with penicillin/streptomycin. The SUM-159 cell line was produced by Professor S. Ethier and were cultured in F-12 HAM's medium supplemented with 5% FBS, 1 μM L-Glutamine, 1 μg ml$^{-1}$ hydrocortisone (BD BioScience, San Diego, CA, USA) and 5 μg ml$^{-1}$ insulin (Novo Nordisk A/S, Målöv, Denmark) and penicillin/streptomycin. The cell lines were routinely tested and found negative for mycoplasma. Conditioned medium from all cell lines was collected at subconfluency and human primary monocytes were cultured for 7–14 days in the conditioned medium (OptiMEM supplemented with 1% penicillin/streptomycin or RPMI-1640 supplemented with 10% FBS and penicillin/streptomycin) or using only medium with 10 ng ml$^{-1}$ rhGM-CSF as control. Human primary M2 macrophages were cultured in OptiMEM supplemented with penicillin/streptomycin and 10 ng ml$^{-1}$ rhGM-CSF for 5 days, after which 20 ng ml$^{-1}$ rhIL-4 was added for another 2 days. All media and supplements were purchased from Thermo Scientific (Logan, UT, USA).

**TMA and immunohistochemistry.** The breast cancer cohort consists of 144 patients diagnosed with invasive breast cancer at Skåne University Hospital, Malmö, Sweden, between 2001 and 2002. The cohort and TMA have previously been described in detail[19,52,53]. CXCL16 cytoplasmic expression was estimated in the malignant cells (intensity: 0 = negative, 1 = weak, 2 = moderate, 3 = strong intensity) and in the stromal fibroblasts (intensity 0 = negative-weak, 1 = strong intensity). Sections (4 μm thick) of the paraffin embedded tumours were mounted onto glass slides and deparaffinized, prior to antigen retrieval using the PT-link system (DAKO, Glostrup, Denmark) and staining in a Autostainer Plus (DAKO) with the EnVisionFlex High pH-kit (DAKO). All histological sections were counterstained with HE. All primary antibodies used for IHC are shown in Supplementary Table 4 (specificity; clone; dilution; company). Sirius Red staining was performed using in house methods. Cytospins were prepared from monocytes or non-enzymatic cell dissociation buffer-collected (Sigma Aldrich) M2 cultures that were air-dried.

**Cytokines and other reagents.** All recombinant human cytokines were obtained from R&D Systems, and the following concentrations were used in all experiments: 10 ng ml$^{-1}$ GM-CSF, 20 ng ml$^{-1}$ IL-4, 100 ng ml$^{-1}$ CXCL12 and 100 ng ml$^{-1}$ CXCL16. For flow cytometry, the following reagents were used (all from BD Biosciences): CD14 clone M5E2, HLA-DR clone G46-6, CD90, EpCAM, CD11b, CD34 annexin V and 7AAD.

The human angiogenesis array Proteome Profiler kit (R&D systems) was used to analyse the soluble mediators according to the manufacturer's protocol. Analysis of CXCL16 in supernatants collected from primary human CAFs was performed using a human specific CXCL16 ELISA kit (R&D Systems) according to the manufacturer's instructions.

**CAF isolation.** Primary human tumours were dissected, minced into smaller pieces and treated overnight with collagenase at 37 °C. The next day, single cell suspensions were prepared and subsequently seeded into large flasks containing HAM/F-12 medium supplemented with 10% FBS and penicillin/streptomycin. After 12 h the non-adherent cells were washed away and the adherent cells were continuously cultured in HAM/F-12 medium supplemented with 10% FBS and penicillin/streptomycin. The medium was changed every other day to remove non-adherent cells until large groups of fibroblasts became apparent. Collected supernatants were stored at −80 °C.

**Boyden chamber migration assays.** Human primary monocytes or M2 macrophages were allowed to migrate through Costar Transwell Permeable Support 8.0 μm (pore size) 24-well plates (Corning; Sigma Aldrich) towards the conditioned medium of breast cancer cells (cultured under serum free conditions), or towards the chemokines CXCL12 or CXCL16 (100 ng ml$^{-1}$).

**Scratch wound assays.** For the scratch wound assays, freshly prepared primary mouse fibroblasts were seeded directly into 6-well plates. Once confluent, a pipette tip was used to scratch the fibroblast monolayer with a total of two scratches per well. To start the assay, the wells were rinsed and conditioned medium applied. The area of the open wound was analysed after 24 h at 37 °C. Cells were collected for quantitative real-time PCR (RT-QPCR) and supernatants were collected for ELISA analyses.

**Quantitative real-time PCR.** Total RNA was extracted according to the manufacturers' instructions using RNeasy Plus kit (Qiagen, Hilden, MD, USA) for fibroblasts or Trizol (Invitrogen, Thermo Scientific) for monocytes. Random hexamers and the M-MuLV reverse transcriptase enzyme (Thermo Scientific) were used and RT-QPCR was performed in triplicate using Maxima SYBR Green/Rox (Thermo Scientific) and analysed on the Mx3005P QPCR system (Agilent Technologies). The relative mRNA expression was normalized to *ACTB*, *HPRT* and *GAPDH* and calculated using the comparative Ct method[54]. For primers see Supplementary Table 5.

**Thymidine incorporation.** Primary human monocytes or primary mouse fibroblasts were cultured in different breast cancer conditioned media and allowed to proliferate for 24 h. [methyl-$^3$H] thymidine (1 μCi) was added for 18 h, and incorporation was determined in a Microbeta Counter (Perkin & Elmer; MA, USA).

**Gene expression profile analysis.** TCGA breast cancer RNAseq data (http://cancergenome.nih.gov/) were downloaded on 30 January 2015. Data were log2 transformed following addition of 1 to each value using R (3.1.1). Breast cancer subtypes were classified using the PAM50 centroids[55] after centring the data around the median.

Gene expression data for the syngeneic tumours were obtained from the previous analysis described in Johnstone et al.[25] and deposited into the Gene Expression Omnibus (G.E.O.) with Accession No. GSE42272 (http://www.ncbi.nlm.nih.gov/geo/).

**Statistical analyses.** Statistics by non-parametric Mann–Whitney U Wilcoxon test, ANOVA (multiple comparisons when indicated in the Figure legends) or student t-test as indicated. SPSS, Graph Pad Prism or R (3.1.1) software was used for statistical analyses.

**Data availability.** The TCGA breast cancer RNAseq data referenced during the study are available in a public repository from the TCGA website (http://cancergenome.nih.gov/). Gene expression data for the syngeneic tumours were obtained from the previous analysis described in Johnstone et al.[25] and deposited into the Gene Expression Omnibus (G.E.O.) with Accession No. GSE42272 (http://www.ncbi.nlm.nih.gov/geo/). All the other supporting the findings of this study are available within the article and its Supplementary Information Files or from the corresponding author (K.L.) upon request.

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

## Acknowledgements

The results are in part based upon data generated by the TCGA Research Network: http://cancergenome.nih.gov/. We thank Ms Elise Nilsson and Mrs Kristina Ekström-Holka for professional technical skills in preparation of IHC. We also thank Drs Susan Evans Axelsson, Kristofer Ahlqvist and Giacomo Canesin for extra help in the animal house. This work was generously supported by grants from the Swedish Research Council, The Swedish Cancer Society, Kocks Foundation, Österlunds Foundation, Gunnar Nilsson Cancer Foundation, MAS Cancer Foundation, Åke Wibergs Foundation, The National Cancer Institute, USA (R01 CA180039 and U01 CA199315 to Z.W.) and finally the SCAN-B project (http://www.med.lu.se/scan_b) and the Mrs Berta Kamprad foundation. Elevate Scientific, www.elevatescientific.com, Lund, Sweden, was used for linguistic corrections and improvements.

## Author contributions

R.A. performed the majority of experiments. C.B., C.H. and S.M. performed the xenograft NSG in vivo experiments and took part in designing the study initially together with K.L. D.B. and S.P. were responsible for the design and execution of in vivo experiments. As clinical pathologists, B.T. and M.E.J. were responsible for the fresh human tumour material and for guiding the scoring of the IHC and verifying all histological stains. C.H., K.S. and Z.W. were responsible for the PDX models. R.L.A. was responsible for the syngeneic models. S.P.E. was responsible for the SUM-159 cells and for critical scientific input. K.J. was responsible for the clinical breast cancer tumour tissue microarray and analyses. C.L. was responsible for analysing RNAseq data, for significant input in both data analysis and in writing the manuscript. K.L. designed all the experiments, wrote the manuscript, as well as interpreted and analysed the data. All authors read and approved the manuscript and all were involved in revising the manuscript.

## Additional information

**Competing financial interests:** The authors declare no competing financial interests.

