## [Peer review file · Nature Communications]

Reviewers' comments:

Reviewer #1: Tumor microenvironment and CAFs
(Remarks to the Author):

Authors focused on the microenvironment of triple negative breast cancer. They examined the xenograft (human cancer cells+ human monocytes) and found that co-implanted human monocytes developed into immunosuppressive myeloid cells. These altered myeloid cells induced activation of stromal fibroblasts and expression of CXCL16 from fibroblasts. These activated fibroblasts can recruit more myeloid cells and fibroblasts. The hypothesis and concept of this article is somewhat interesting and novel, however serious problems are raised.

Major points

1) Why did authors use human breast cancer cell lines and human monocytes as xenograft? Human breast carcinoma is acceptable for experiments, however, co-transplantation of human monocytes is of no value in this experiment. Why didn't authors analyze the host (mouse)-derived stromal cells? Authors can examine the immunophenotype of recruited stromal cells by using mouse-specific antibodies. Reviewer thinks this experiment is artificial.

2) Authors mentioned that co-transplanted human monocytes changed their phenotype to myeloid derived suppressor cells. However these cells will be changed into mouse-derived stromal cells.

3) In xenograft model. Authors analyzed only MDA-MB-231 cells as TN and MCF-7 as non-TN. Authors should examine using one or more cell lines.

4) The antibodies used here are human or mouse specific antibody? This information should be mentioned.

5) Page 8. Authors TN tumor environment induced survival and proliferation of co-transplanted myeloid cells. Why didn't authors check the S100A9 expression? Author should indicate the differentiation-related markers.

6) About soluble mediators. Authors indicated the results of cancer cell line supernatants. However reviewer wonders whether these sup. contained both cancer cell-derived and monocytes-derived factors. How did authors collect the sup. only from cancer cells?

7) page 12. Authors examined the frequency of apoptosis (annexin V positive cells) of fibroblasts. For what ?

Minor points

1) Figure 1. S100A9 expression in MDA-MB-231 + Monocytes. The color of this positive reaction is obviously differ from others. Reviewer wonders this reaction is non-specific reaction.

2) Figure 4. Collagen VI expression is found in the cytoplasm. The location of collagen VI should be also in the extracellular spaces.

Reviewer #2: Tumor immunoenvironment and monocytes
(Remarks to the Author):

The paper of Allaoui, Leandersson and colleagues reports on the role for CXCL16 in monocyte attraction and resulting stroma activation in the setting of triple-negative breast cancer (TNBC). The manuscript is well-prepared and the currently presented data is convincing. The results may

shift the current paradigm that CXCL16 is a chemokine preferentially promoting attraction of antitumor effector cells (such as CTLs and NK cells). Addressing the following issues would allow the authors to demonstrate the general applicability and physiologic relevance of the central message:

- 1) The current concept of the unique contribution of CXCL16-driven events to pathogens of TNBC seems to be solely based on the comparison of four BC cell lines: MCF-7 and T47D (Luminal A), compared to MDA-MD-231 and MDA-MD-468 (TNBC). While the presented results are indeed intriguing, data from additional cell lines and, ideally, from primary tumor material would allow to extrapolate these results to TNBC in general.
- 2) The proposed difference with the previously-accepted paradigm needs additional discussion and explanation. In particular, I miss the experiments allowing the authors to exclude the possibility that the currently described paradoxical effects reflect the mismatch between the human and mouse component of the NSG-xenotransplant model. Studies in an *in vivo* all human model and /or all mouse models would help to eliminate that possibility and further support the central message of the paper. The current manuscript contains data from co-cultures of human monocytes and cancer cells, but the impact on human fibroblasts is not studied.
- 3) The central message of the paper and physiologic relevance of the reported observations could be further supported by the demonstration of correlations between CXCL16 expression and markers of activated macrophages and fibroblasts in clinical samples.

Minor:

- 1) The abstract would benefit from additional edits for clarity. The current version makes it difficult to appreciate the proposed sequence of events and their causal relations.

Reviewer #3: Breast cancer microenvironment
(Remarks to the Author):

These authors have investigated differences in monocyte-tumor cell interactions in xenografts that may lead to a change in the recruitment of fibroblasts. Both xenografts and conditioned media experiments are utilized to show that the interactions observed *in vivo* are reproducible in a culture system. Boyden chamber assays were used to confirm the migration to TN (and not LumA) breast cancers. The collection of data implicates CXCL16 in activation of the microenvironment in basal-like or triple negative breast cancers. The manuscript investigates an original hypothesis and provides a large range of data to evaluate the hypothesis very thoroughly. Most conclusions are justified by the data. There are a few omissions which should be addressed and a few points needing improved clarity or support from previous literature.

The immunohistochemistry marker information is shown with representative pictures, and no quantitation. The authors should consider semi-quantitative scoring and/or automated image analysis for more quantitative results. It is difficult to judge whether the qualitative changes are reproducible across replicates with only a small number of xenografts (and small areas within them) pictured.

Many of the phenotypic assays use only MCF7 cells and 231 cells. The authors began with four cell lines and should comment on the reproducibility of the phenotypes across other cell lines. It is expected that the experiments may have shown variance - this has been reported in other studies of cocultures with triple negative cell lines. While this would not undermine the main findings, it is important to report.

Some authors have argued that the 231 cells may be claudin-low rather than basal-like. They are a very mesenchymal-like cancer cell line. Is it this characteristic that distinguishes their response from that of macrophages? Are they already secreting profiles similar to cancer-associated fibroblasts? This is related to the previous comment about reproducibility across cell lines. If not all TNs exhibit this characteristic, are there phenotypes of the TNs that do?

The authors do not reference any of the previous literature on cocultures of these cell lines. They do make not of some prior work by Su et al. in the text, but the reference for this is not matched

to the number at the end of the sentence. There is no paper by Su et al. cited. The authors should more comprehensively evaluate the literature on cocultures of macrophages and breast cancer cells to evaluate whether there is existing complementary data. A brief search of the terms 'macrophage' and 'basal-like' and 'coculture' suggests that some previous similar studies have been conducted, but these previous data have not been considered herein.

Minor comments

On page 8, the authors argue that cytokines from luminal A cultures were less 'invasive' in character. This terminology is unclear and should be replaced with something more specific.

Unclear sentence in introduction: "It is known since before that both monocytes and BMDCs can induce metastasis to distant sites"

Reviewers' comments:

Reviewer #1: Tumor microenvironment and CAFs
(Remarks to the Author):

Authors focused on the microenvironment of triple negative breast cancer. They examined the xenograft (human cancer cells+ human monocytes) and found that co-implanted human monocytes developed into immunosuppressive myeloid cells. These altered myeloid cells induced activation of stromal fibroblasts and expression of CXCL16 from fibroblasts. These activated fibroblasts can recruit more myeloid cells and fibroblasts. The hypothesis and concept of this article is somewhat interesting and novel, however serious problems are raised.

Major points

1) Why did authors use human breast cancer cell lines and human monocytes as xenograft? Human breast carcinoma is acceptable for experiments, however, co-transplantation of human monocytes is of no value in this experiment. Why didn't authors analyze the host (mouse)-derived

stromal cells? Authors can examine the immunophenotype of recruited stromal cells by using mouse-specific antibodies. Reviewer thinks this experiment is artificial.

The aim initially was indeed to use human myeloid cells and human cancer cells, to inject them together, to eliminate the factor of different infiltration rates in the different tumor types. From our own experience, we also believed that human myeloid cells probably would respond better to with human cancer cells (as in contrast to mouse myeloid and human cancer cells), an issue that had not been experimented on in vivo before. We wanted to ask whether human myeloid cells affect stroma formation differently in luminal versus TN human breast cancer, i) with the same amount of myeloid cells present, ii) in the same location in the tumors, iii) from the very first day. We do respect reviewer 1s opinion on the artificial setting however. We have therefore now added new physiologically relevant preclinical models to investigate this:

- syngeneic mouse breast tumor models using a luminal (67NR) tumors and a TN (4T1) tumor model in Balb/C immunocompetent mice. In this model we analyze the host (mouse)-derived stromal and myeloid cells in the mouse derived graft. Supplementary Figure 5 and Supplementary Table 3.
- patient derived xenograft models (PDX) using luminal and TN human breast tumor grafts in Nod-SCID mice. In this model we analyze the host (mouse)-derived stromal and myeloid cells in the patient derived graft. Figure 4 and Figure 6G.

In both models (PDX and syngeneic) we see that:

- More myeloid cells are present in the TN grafts
- The infiltrating myeloid cells express immunosuppressive markers (S100A9) significantly more in the TN grafts
- Activated fibroblasts (α SMA expressing) are significantly more pronounced in the TN grafts
- Mouse derived CXCL16 is significantly upregulated in the TN PDX grafts as well as the TN syngeneic grafts.

Immunocompetent BALB/c, NOD-Scid and NSG-mice all have myeloid cells (although with different grades of maturity), and therefore the question as to how they affect the stroma formation is similar but with the difference that fully competent human myeloid cells are grafted in the NSG-mice as well. The results from the different models are similar, but significantly more pronounced in the models using grafted human myeloid cells, a finding that is novel and important for the interpretation of different models. We also want to take the opportunity to emphasize that the α SMA antibody used for activated stromal cells recognize both human and mouse origin and in all the xenograft models performed, also in the original manuscript file, we state that most cells expressing α SMA are of mouse origin. This is also shown using double stainings in the manuscript (Fig S5A). The stromal fibroblast cells in our models therefore actually come from mice. The specificity of antibodies are now added as a separate Supplementary Table 4 and referred to in the material and methods sections for all antibodies used. All antibodies have

been tested and titrated on both mouse and human tissue. We strongly believe that mixing in human myeloid cells together with the human tumor cells led to clearer results when it comes to what the myeloid cells actually do in the tumor, since infiltration and migration will have an inferior role when it comes to drawing conclusions.

2) Authors mentioned that co-transplanted human monocytes changed their phenotype to myeloid derived suppressor cells. However these cells will be changed into mouse-derived stromal cells.

The S100A9 antibodies used in this manuscript are strictly species-specific (See New Supplementary Table 4 for all antibody specificities). Thus, anti-human S100A9 stainings only detect the grafted cells. Using double stainings (Figure 2A and Supplementary Figure 5A) we have also shown that:

a) the human myeloid cells are the ones that express S100A9, keeping their human myeloid markers

b) that most α SMA⁺ cells are of mouse origin (the α SMA antibody recognizes both human and mouse). The human myeloid cells are definitely part of the tumor stroma, but they do not trans-differentiate or change into mouse stromal cells (Figure 2A and Supplementary Figure 5A).

The CD11b, CD163 and S100A9 antibodies only recognize human origin. The α SMA antibodies recognize both human and mouse origin. This is now added as a separate Supplementary Table 4 and referred to in the material and methods sections for all antibodies used. All antibodies have been tested and titrated on both mouse and human tissue.

3) In xenograft model. Authors analyzed only MDA-MB-231 cells as TN and MCF-7 as non-TN. Authors should examine using one or more cell lines.

We have repeated all experiments using one more luminal (T47D) and two more TN (MDA-MB-468 and SUM-159) cells. The xenografts were also repeated using one more luminal (T47D) and one more TN (SUM-159) cells. These were chosen since they do not express S100A9 endogenously in the malignant cells as MDA-MB-468 cells do (presented in our publication last year; Bergenfelz et al Br J Cancer 2015). New data in Figures 2, 3, 5, 6, S1, S4, S7.

4) The antibodies used here are human or mouse specific antibody? This information should be mentioned.

This is now added as a separate Supplementary Table 4 and referred to in the material and methods sections for all antibodies used. All antibodies have been tested and titrated on both mouse and human tissue.

5) Page 8. Authors TN tumor environment induced survival and proliferation of co-transplanted myeloid cells. Why didn't authors check the S100A9 expression? Author should indicate the differentiation-related markers.

The differentiation markers are checked (shown in Supplementary Figure 2;

former Supplementary Figure 1).

S100A9 mRNA expression was analyzed and we are unable to see an upregulation in vitro. In the syngeneic models we do see a significant upregulation in the TN grafts (Figure S5 and Table S3). In the PDX-models, we also do see an upregulation at the mRNA level in a TN graft, indicating that in vivo settings are needed to see this. S100A9 is known to be regulated at the post-transcriptional level and this might explain our findings. We have added a sentence about this in the results pg 12.

6) About soluble mediators. Authors indicated the results of cancer cell line supernatants. However reviewer wonders whether these sup. contained both cancer cell-derived and monocytes-derived factors. How did authors collect the sup. only from cancer cells?

The medium used when preparing the supernatant was serum-free and had previously been carefully chosen to affect the differentiation of myeloid cells minimally (OptiMeM; Bergenfelz et al J Immunol 2012) as indicated in the materials and methods. These supernatants were collected after culture for 7 days of the indicated breast cancer cell lines and harvested at subconfluency, or cultured with primary monocytes added (co-cultures) for the same amount of days (as indicated in Materials and methods). Hence, the same amount of days were used for only cell-lines as for the co-cultures. We have also added new data to this series from TN SUM-159 and TN SUM-159 / monocytes now (Supplementary Figure 4).

7) page 12. Authors examined the frequency of apoptosis (annexin V positive cells) of fibroblasts. For what?

We wanted to investigate all three possibilities of the increased proportion of fibroblasts: 1) migration/infiltration 2) proliferation 3) survival.

Minor points

1) Figure 1. S100A9 expression in MDA-MB-231+Monocytes. The color of this positive reaction is obviously different from others. Reviewer wonders this reaction is non-specific reaction.

The pattern of S100A9 stainings can vary, sometimes seen as cloudy stainings (mostly in mouse) and sometimes as distinct. The anti-human S100A9 antibody used here has been thoroughly tested on human breast tissue with expertise from Medical Pathologists from the Department of Clinical Pathology, and was published last year (Bergenfelz et al B J Cancer 2105). In this publication the antibody was tested rigorously. We now also confirm the staining using both the MDA-MB-231 / monocyte (Fig 1) as well as the SUM-159 / monocyte xenografts for this revision (Supplementary Figure 1). All antibodies have been tested and titrated on both mouse and human tissue.

2) Figure 4. Collagen VI expression is found in the cytoplasm. The location of collagen VI should be also in the extracellular spaces.

We believe that this is a matter of image resolution and hope that the full size images will show this better. It is indeed present in the extracellular spaces as well.

Reviewer #2: Tumor immunoenvironment and monocytes
(Remarks to the Author):

The paper of Allaoui, Leandersson and colleagues reports on the role for CXCL16 in monocyte attraction and resulting stroma activation in the setting of triple-negative breast cancer (TNBC). The manuscript is well-prepared and the currently presented data is convincing. The results may shift the current paradigm that CXCL16 is a chemokine preferentially promoting attraction of antitumor effector cells (such as CTLs and NK cells). Addressing the following issues would allow the authors to demonstrate the general applicability and physiologic relevance of the central message:

- 1) The current concept of the unique contribution of CXCL16-driven events to pathogenesis of TNBC seems to be solely based on the comparison of four BC cell lines: MCF-7 and T47D (Luminal A), compared to MDA-MD-231 and MDA-MD-468 (TNBC). While the presented results are indeed intriguing, data from additional cell lines and, ideally, from primary tumor material would allow to extrapolate these results to TNBC in general.

We appreciate this comment from Reviewer 2. We have now extended our experiments to involve:

- Repeated all *in vitro* experiments using three more cell lines (T47D, MDA-MB-468 and the novel TN SUM-159).
- Repeated the xenograft models using more cell lines (T47D and the TN SUM-159 in collaboration with Professor Stephen P Ethier, South Carolina, USA)
- Added analysis of a human breast cancer clinical tumor tissue microarray in collaboration with Clinical Pathologist Professor Karin Jirström, Lund, Sweden
- Added preclinical PDX models. We have performed luminal and TN breast cancer patient derived xenografts (PDX) in collaboration with Professor Zena Werb's lab in San Francisco, USA
- Added preclinical syngeneic models. We have performed syngeneic luminal vs TN mouse mammary tumor models in collaboration with Professor Robin Anderson, Melbourne, Australia

All new data are in line with our previous findings and supporting our conclusions. The results from the tissue microarray analyzed further supports our conclusions, but also gives a better extrapolation to TNBC in general.

- 2) The proposed difference with the previously-accepted paradigm needs additional discussion and explanation. In particular, I miss the

experiments allowing the authors to exclude the possibility that the currently described paradoxical effects reflect the mismatch between the human and mouse component of the NSG-xenotransplant model. Studies in an and ex vivo all human model and /or all mouse models would help to eliminate that possibility and further support the central message of the paper. The current manuscript contains data from co-cultures of human monocytes and cancer cells, but the impact on human fibroblasts is not studied.

We appreciate this comment. We have therefore performed new experiments using both PDX and syngeneic immunocompetent models. Please see answers to point 1 above. We have:

- Added preclinical PDX models. We have performed luminal and TN breast cancer patient derived xenografts (PDX) in collaboration with Professor Zena Werb's lab in San Francisco, USA
- Added preclinical syngeneic models. We have performed syngeneic luminal vs TN mouse mammary tumor models in collaboration with Professor Robin Anderson, Melbourne, Australia
- Repeated the xenograft models using more cell lines (T47D and the TN SUM-159)

All new data are in line with our previous findings, thus supporting our conclusions.

The impact on human fibroblasts is indeed interesting. We believe that when the human primary cancer associated fibroblasts are isolated from tumors they are already activated. We have therefore instead analyzed the expression pattern of both CXCL16 and Collagen depositions in the fibroblasts, in different breast tumors, using a human breast cancer cohort consisting of 144 patients. We find that a high level of CXCL16 expression in the fibroblasts *per se* correlates significantly with TN breast cancers. We also find that collagen depositions of Collagens I and III are lower specifically in TN breast tumors. These tumors all have macrophages present in the tumor stroma. We have also repeated all *in vitro* fibroblast experiments using more cell lines. We sincerely hope that reviewer 2 finds these data appropriate. New Table 1. Fig 5. New Fig S7. Please also see point 3 below.

- 3) The central message of the paper and physiologic relevance of the reported observations could be further supported by the demonstration of correlations between CXCL16 expression and markers of activated macrophages and fibroblasts in clinical samples.

We thank reviewer 2 for this suggestion.

- We have now analyzed the expression pattern of both CXCL16 and Collagen depositions in cancer associated fibroblasts in a human breast cancer tissue microarray cohort consisting of 144 patients and compared it to clinical parameters. We find that a high fibroblast expression of CXCL16 as well as low Collagen I / III depositions, correlate significantly with TN breast cancers. These tumors all have

macrophages present in the tumor stroma (also see our previous publication Medrek et al The presence of tumor associated macrophages in tumor stroma as a prognostic marker for breast cancer patients, BMC Cancer 2012).

- This is in line with our novel data from the PDX models, where CXCL16 originating from mouse cells, is upregulated in TN PDX tumors only. Indeed, also the syngeneic models show that CXCL16 is expressed at significantly higher levels in the TN 4T1 tumors as compare to the luminal 67NR tumors. In these models, also aSMA is expressed at significantly higher levels in the fibroblasts in the TN tumors. New Table 1, New Fig 4, New Fig 6G and New Supplementary Fig 5.
- We have also performed more experiments on the migration of M2 macrophages towards CXCL16 and this experiment is now significant.
- Finally, we have analyzed the expression levels of the receptor for CXCL16 (CXCR6) on primary human monocytes and could show that indeed, this receptor is expressed on a fraction of monocytes (data not shown; although we would be happy to include these data upon request).

Minor:

The abstract would benefit from additional edits for clarity. The current version makes it difficult to appreciate the proposed sequence of events and their causal relations.

We apologize for this. The abstract has been looked over and a sentence has been added. It still meets the criteria of less than 150 words.

Reviewer #3: Breast cancer microenvironment
(Remarks to the Author):

These authors have investigated differences in monocyte-tumor cell interactions in xenografts that may lead to a change in the recruitment of fibroblasts. Both xenografts and conditioned media experiments are utilized to show that the interactions observed in vivo are reproducible in a culture system. Boyden chamber assays were used to confirm the migration to TN (and not LumA) breast cancers. The collection of data implicates CXCL16 in activation of the microenvironment in basal-like or triple negative breast cancers. The manuscript investigates an original hypothesis and provides a large range of data to evaluate the hypothesis very thoroughly. Most conclusions are justified by the data. There are a few omissions which should be addressed and a few points needing improved clarity or support from previous literature.

The immunohistochemistry marker information is shown with representative pictures, and no quantitation. The authors should consider semi-quantitative scoring and/or automated image analysis for more quantitative results. It is difficult to judge whether the qualitative changes are reproducible across

replicates with only a small number of xenografts (and small areas within them) pictured.

Thank you for this comment. We have now added histograms with statistical evaluation of the IHC quantifications for each protein (Fig 1, Fig 2, Supplementary Fig 1) and marked this more carefully in Supplementary Table 1-2. We have gone through the figures and believe that the magnifications used are the most appropriate, showing a large part of the tumor, but still with a good magnification. Instead we have increased the file-size slightly. We hope that the quantifications and improvements done will help in judging the stainings and results better.

Many of the phenotypic assays use only MCF7 cells and 231 cells. The authors began with four cell lines and should comment on the reproducibility of the phenotypes across other cell lines. It is expected that the experiments may have shown variance - this has been reported in other studies of cocultures with triple negative cell lines. While this would not undermine the main findings, it is important to report.

We appreciate this comment. We have now extended our experiments to involve:

- Repeated all *in vitro* experiments using three more cell lines (T47D, MDA-MB-468 and the novel TN SUM-159).
- Repeated the xenograft models using more cell lines (T47D and the TN SUM-159)
- Added analysis of a human breast cancer clinical tumor tissue microarray in collaboration with Clinical Pathologist Professor Karin Jirström, Lund, Sweden
- Added preclinical PDX models. We have performed luminal and TN breast cancer patient derived xenografts (PDX) in collaboration with Professor Zena Werb's lab in San Francisco, USA
- Added preclinical syngeneic models. We have performed syngeneic luminal vs TN mouse mammary tumor models in collaboration with Professor Robin Anderson, Melbourne, Australia

All new data are in line with our previous findings and supporting our conclusions.

Some authors have argued that the 231 cells may be claudin-low rather than basal-like. They are a very mesenchymal-like cancer cell line. Is it this characteristic that distinguishes their response from that of macrophages? Are they already secreting profiles similar to cancer-associated fibroblasts? This is related to the previous comment about reproducibility across cell lines. If not all TNs exhibit this characteristic, are there phenotypes of the TNs that do?

This is a very important point. We have therefore included three TN cell lines now. Most of the new data from the new TN cell lines MDA-MB-468 and SUM-159 cells are in line with what we showed in the first draft (using MDA-MB-231), although some of the *in vitro* data do differ between TN cell lines (e.g., monocyte and fibroblast proliferation). Most importantly however, the *in vivo* Xenograft models show the same result concerning differentiation of myeloid cells into S100A9+ immunosuppressive cells and activation of fibroblasts (α SMA⁺) only in a TN environment, using either MDA-MB-231 or SUM-159 cells (Figure 1-2, Supplementary Figure 1 and Supplementary Table 1-2), and

the four PDX tumors from different patients show increased CXCL16 expression (Figure 4 and Figure 6G). In the TCGA-database CXCL16 correlate to the basal breast cancer subgroup (Figure 7A). Finally, in the human breast cancer clinical tumor tissue microarray we found that CXCL16 was expressed at high levels in TN fibroblasts (Table 1), but we did not find a correlation to any specific TN subtype (data not shown). This does not rule out this possibility and therefore we have added this discussion in the text.

The authors do not reference any of the previous literature on cocultures of these cell lines. They do make not of some prior work by Su et al. in the text, but the reference for this is not matched to the number at the end of the sentence. There is no paper by Su et al. cited.

This is an embarrassing mistake, and was supposed to be there. It probably disappeared during the editing process. We have now gone through the manuscript and added this important reference.

The authors should more comprehensively evaluate the literature on cocultures of macrophages and breast cancer cells to evaluate whether there is existing complementary data. A brief search of the terms 'macrophage' and 'basal-like' and 'coculture' suggests that some previous similar studies have been conducted, but these previous data have not been considered herein.

We have now added and commented on references concerning primary human monocytes/macrophages and TN vs luminal breast cancer specifically in the discussion.

Minor comments

On page 8, the authors argue that cytokines from luminal A cultures were less 'invasive' in character. This terminology is unclear and should be replaced with something more specific.

We agree with this comment and removed the word invasive. It now says "the luminal A breast cancer cells (blue box) expressed fewer factors".

Unclear sentence in introduction: "It is known since before that both monocytes and BMDCs can induce metastasis to distant sites"

We appreciate this comment and have now changed the sentence. We hope that it is more direct this way.

REVIEWERS' COMMENTS:

Reviewer #1 (Remarks to the Author):

The authors have responded adequately to the major concerns raised in the initial review.

Reviewer #2 (Remarks to the Author):

A convincing response to the critiques and a thorough revision of the paper. I do not have any additional comments on this interesting paper.

Reviewer #3 (Remarks to the Author):

The authors have added substantial new data and strengthened the discussion. A few minor points related to references and clarity were also well-addressed.